# The tyrosine phosphatases LAR and PTPRδ act as receptors of the nidogen-tetanus toxin complex

Sunaina Surana [iD] [1,2,3 ✉], David Villarroel-Campos [iD] [1,2,3], Elena R Rhymes[1,2], Maria Kalyukina[4], Chiara Panzi [iD] [1,2,3], Sergey S Novoselov [iD] [1,2], Federico Fabris[5], Sandy Richter[1,5], Marco Pirazzini[5], Giuseppe Zanotti[5], James N Sleigh [iD] [1,2,3] & Giampietro Schiavo [iD] [1,2,3 ✉]

## Abstract

**Tetanus neurotoxin (TeNT) causes spastic paralysis by inhibiting neurotransmission in spinal inhibitory interneurons. TeNT binds to the neuromuscular junction, leading to its internalisation into motor neurons and subsequent transcytosis into interneurons. While the extracellular matrix proteins nidogens are essential for TeNT binding, the molecular composition of its receptor complex remains unclear. Here, we show that the receptor-type protein tyrosine phosphatases LAR and PTPRδ interact with the nidogen-TeNT complex, enabling its neuronal uptake. Binding of LAR and PTPRδ to the toxin complex is mediated by their immunoglobulin and fibronectin III domains, which we harnessed to inhibit TeNT entry into motor neurons and protect mice from TeNT-induced paralysis. This function of LAR is independent of its role in regulating TrkB receptor activity, which augments axonal transport of TeNT. These findings reveal a multi-subunit receptor complex for TeNT and demonstrate a novel trafficking route for extracellular matrix proteins. Our study offers potential new avenues for developing therapeutics to prevent tetanus and dissecting the mechanisms controlling the targeting of physiological ligands to long-distance axonal transport in the nervous system.**

**Keywords** LAR; Neuromuscular Junction; Nidogen; PTPRδ; Tetanus Toxin
**Subject Categories** Microbiology, Virology & Host Pathogen Interaction; Neuroscience

## Introduction

Tetanus neurotoxin (TeNT) is one of the most toxic molecules known. Produced by the anaerobic, Gram-positive bacterium *Clostridium tetani*, TeNT causes tetanus, a neuroparalytic syndrome characterised by lockjaw, opisthotonus, muscle stiffness and increasingly painful spasms, ultimately leading to respiratory failure and death (Farrar et al, 2000). Despite the availability of an effective vaccine and antitoxin preparations, tetanus is a leading cause of mortality in neonates and unvaccinated adults in developing countries due to limited resources, lack of enforcement of appropriate public healthcare measures and scarce availability of treatments (Thwaites et al, 2015; Pirazzini et al, 2022).

TeNT is formed by three modular domains, each of which is essential for intoxication of the nervous system. In the active toxin, these domains form a heavy chain (H chain) and a catalytic light chain (L chain). Generated from a single polypeptide by proteolytic cleavage, both subunits remain associated *via* non-covalent interactions and a conserved inter-chain disulfide bond. The H chain is further subdivided into two domains: an amino terminal ($H_N$) and a carboxy terminal ($H_C$) domain, which are responsible for membrane translocation and receptor binding, respectively (Schiavo et al, 2000; Surana et al, 2018). After bacterial spore germination and TeNT production, the active toxin accumulates in the synaptic space at the neuromuscular junction (NMJ) and binds to the plasma membrane of motor neurons with sub-nanomolar affinity by virtue of its $H_C$ domain (Binz and Rummel, 2009). This results in rapid internalisation of the neurotoxin, followed by its long-distance, retrograde transport towards the neuronal cell body in the spinal cord (Salinas et al, 2010). TeNT then undergoes trans-synaptic transfer into inhibitory interneurons, where the $H_N$ domain drives the pH-dependent translocation of the L chain from the endocytic lumen of synaptic vesicles into the cytosol (Pirazzini et al, 2016). The L chain, which possesses zinc protease activity, cleaves the synaptic vesicle protein VAMP/synaptobrevin, leading to a cessation of neurotransmitter release (Schiavo et al, 1992, 1994). This perturbs the balance of excitatory and inhibitory inputs to motor neurons, leading to motor neuron hyperactivity and spastic paralysis (Schiavo et al, 2000; Surana et al, 2018).

Given the high neuro-specificity and extreme toxicity of TeNT, several studies have endeavoured to identify the presynaptic receptors responsible for its entry into motor neuron terminals. The C-terminal region of the $H_C$ domain ($H_{CC}$) was found to bind with high affinity to polysialogangliosides of the G1b subgroup (Chen et al, 2009). This interaction, which takes place *via* two sialic acid-binding pockets, is essential for TeNT intoxication since site-

[1]Department of Neuromuscular Diseases, Queen Square Institute of Neurology, University College London, London WC1N 3BG, UK. [2]UCL Queen Square Motor Neuron Disease Centre, University College London, London WC1N 3BG, UK. [3]UK Dementia Research Institute, University College London, London WC1E 6BT, UK. [4]Department of Clinical and Experimental Epilepsy, Queen Square Institute of Neurology, University College London, London WC1N 3BG, UK. [5]Department of Biomedical Sciences, University of Padova, Padova 35131, Italy. ✉E-mail: s.surana@ucl.ac.uk; giampietro.schiavo@ucl.ac.uk

directed mutagenesis of key residues within these sites led to a dramatic loss of binding to rat brain synaptosomes as well as abrogation of toxicity in a phrenic nerve-diaphragm preparation (Rummel et al, 2003). However, whereas polysialogangliosides are enriched in the neuronal plasma membrane, they are not exclusively present on the presynaptic surface of motor neurons. In addition, membrane binding of TeNT was found to be protease-sensitive (Pierce et al, 1986; Lazarovici and Yavin, 1986). This, together with the finding that the $H_C$-GT1b interaction displays lower affinity in vitro than that observed for $H_C$ binding to rat brain synaptosomes, suggested that instead of relying solely on polysialogangliosides, TeNT binding to motor neurons also requires a specific membrane protein receptor (Montecucco et al, 2004; Rummel et al, 2003).

Early studies indicated that the protein receptor for TeNT was a glycosylphosphoinositol (GPI)-anchored protein resident in lipid microdomains, since treatment of mouse spinal cord cells with phosphatidylinositol-specific phospholipase C inhibited TeNT-induced cleavage of VAMP/synaptobrevin (Munro et al, 2001). This was supported by the finding that gangliosides accumulate in lipid microdomains (Simons and Toomre, 2000). These observations led to the identification of Thy-1 as a TeNT-binding protein in sphingolipid-enriched membrane regions. However, independent evidence indicated that Thy-1 was not the neuronal receptor for TeNT since $H_C$ binding and internalisation in Thy-1 knockout mice was indistinguishable from wild-type mice (Herreros et al, 2001). More recently, the extracellular matrix (ECM) proteins nidogens (also called entactins) were identified as co-receptors of TeNT at the NMJ. The $H_C$ domain was found to bind preferentially to nidogen-rich regions of the NMJ, and both nidogen-1 and -2 were found to directly interact with TeNT. Furthermore, peptides derived from nidogen-1 inhibited membrane binding and uptake of TeNT in motor neurons, thus preventing the appearance of spastic paralysis in intoxicated mice (Bercsenyi et al, 2014). However, the precise identity of the membrane receptor that engages with the nidogen-TeNT complex and ferries it into motor neurons remains unknown.

Several lines of evidence suggest that the LAR (leukocyte common antigen-related protein) family of tyrosine phosphatases plays a role in the internalisation of TeNT. O'Grady and colleagues have shown that the nidogen-laminin complex interacts with recombinant LAR in vitro, a result supported by the observation that the *Caenorhabditis elegans* orthologue of LAR, *ptp-3*, was found to genetically interact with *nid-1* (O'Grady et al, 1998; Ackley et al, 2005). Furthermore, LAR is sequestered in lipid microdomains on the plasma membrane and co-localises with caveolin-enriched fractions (Caselli et al, 2002). LAR has also been linked to trophic pathways in the nervous system, including in the regulation of BDNF signalling through its receptor TrkB in hippocampal neurons (Yang et al, 2006). Knockdown of the LAR homologue PTPRδ in cortical neurons was found to potentiate phosphorylation of TrkB as well as its downstream effector molecules MEK and ERK (Tomita et al, 2020). Concomitantly, TeNT has been shown to trigger TrkB phosphorylation, activate downstream Akt and ERK signalling and share retrograde signalling endosomes with BDNF, and its receptors TrkB and p75[NTR], during its axonal journey to the spinal cord (Lalli and Schiavo, 2002; Deinhardt et al, 2006; Gil et al, 2003; Calvo et al, 2012). Altogether, this evidence highlights

functional links between the nidogen-TeNT complex and LAR phosphatases.

LAR, along with PTPRδ and PTPRσ, belongs to the type IIa family of transmembrane receptor-type protein tyrosine phosphatases (RPTPs). These proteins contain three extracellular immunoglobulin-like (Ig) domains, followed by four to eight fibronectin III (FNIII) domains, depending on alternative splicing. The intracellular subunit contains two phosphatase domains: D1 and D2, of which only the D1 domain is catalytically active (Fig. 1A). Mature RPTPs undergo constitutive proteolysis between the FNIII and D1 domains, generating an extracellular subunit that is non-covalently bound to the phosphatase domain-containing moiety (Takahashi and Craig, 2013). All three LAR-RPTPs have been shown to regulate cellular adhesion and signalling, thus playing important roles in the nervous system, including synapse formation and stabilisation, neurite outgrowth and axon guidance as well as sprouting and innervation in cholinergic neurons (Cornejo et al, 2021).

In this study, we report that LAR and PTPRδ are components of the nidogen-TeNT receptor complex and enable its internalisation in motor neurons. We show that depletion of LAR is sufficient to abrogate neuronal uptake of the $H_C$ domain of TeNT (henceforth referred to as $H_C$T). Binding of these proteins to the nidogen-TeNT complex is mediated by specific immunoglobulin and fibronectin III domains, which we leveraged as tools to inhibit $H_C$T entry into motor neurons. This function of LAR is independent of its role in regulating the neurotrophic activity of TrkB. Furthermore, abrogation of the interaction between the nidogen-TeNT complex and LAR/PTPRδ protects mice from TeNT-induced spastic paralysis. Taken together, our results define the physiological receptors for TeNT on the neuronal plasma membrane and yield important insights into the mechanisms controlling the binding and internalisation of physiological ligands and toxins, such as TeNT, into the nervous system. Importantly, the identification of these receptor complexes paves the way for developing therapeutics to prevent tetanus.

## Results

### Nidogens interact with the receptor-type protein tyrosine phosphatases LAR and PTPRδ

The starting point of our investigation was the observation that TeNT is critically dependent on nidogens for its binding to mammalian NMJs, together with the previously reported in vitro interaction between LAR and the nidogen-laminin complex (O'Grady et al, 1998; Bercsenyi et al, 2014). While LAR phosphatases are broadly distributed in the nervous system (Pulido et al, 1995; Dunah et al, 2005; Kwon et al, 2010), it is unclear whether all three proteins are specifically expressed in motor neurons. When spinal ventral horn cultures were immunostained for each individual LAR phosphatase, we found that LAR, PTPRδ and PTPRσ were all abundantly present in neuronal cell bodies as well as neurites (Appendix Fig. S1A). When these cultures were co-stained for the mature motor neuron marker choline acetyl transferase (ChAT) (Barber et al, 1984; Sances et al, 2016), we found the presence of all three proteins in ChAT[+] neurons

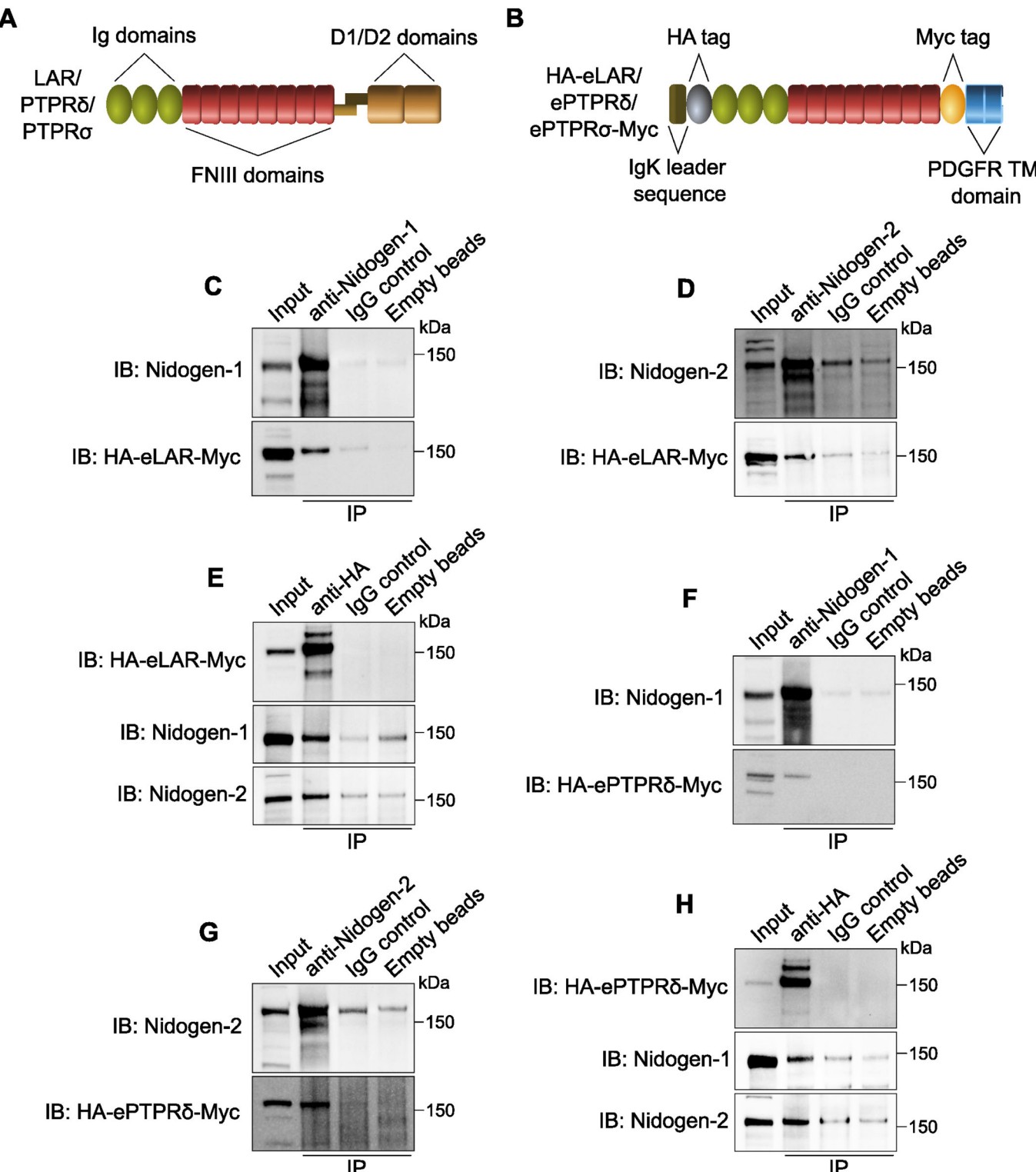

(Appendix Fig. S1A), confirming their expression in spinal cord motor neurons.

For LAR to act as a receptor for TeNT at the NMJ, thus enabling its internalisation into motor neurons, its extracellular domain should be able to bind to the TeNT-nidogen complex. To test this hypothesis, we performed co-immunoprecipitation experiments between the extracellular subunit of LAR (eLAR) and nidogens in the presence of $H_CT$. The Ig and FNIII domains of human LAR were fused to a haemagglutinin (HA) tag at the N-terminus; the C-terminus was fused to a Myc tag and the transmembrane domain

**Figure 1.   Nidogens bind to the receptor-type protein tyrosine phosphatases (RPTPs) LAR and PTPRδ.**

(**A**) Schematic showing the domain organisation of the LAR family of RPTPs. Full-length LAR, PTPRδ and PTPRσ contain three extracellular immunoglobulin-like (Ig) and eight fibronectin III (FNIII) domains, as well as two intracellular protein tyrosine phosphatase (D1 and D2) domains. RPTPs undergo proteolytic processing between the FNIII and phosphatase domains; this generates an extracellular subunit that remains non-covalently bound to the intracellular phosphatase subunit. (**B**) Schematic diagram of human LAR, PTPRδ and PTPRσ chimeric proteins used for co-immunoprecipitations. The extracellular domain of each of these RPTPs was fused to the murine Igκ-chain leader sequence and haemagglutinin (HA) tag at the N-terminus; the C-terminus was fused to the platelet-derived growth factor receptor transmembrane (PDGFR TM) domain and a Myc tag. (**C–E**) Western blots showing the direct interaction of nidogen-1 and -2 with the extracellular domain of LAR in the presence of VSVG-$H_C$T. Nidogen-1 (**C**) and nidogen-2 (**D**) were immunoprecipitated from N2a cell lysates, and co-immunoprecipitates were probed using an anti-HA antibody. Conversely, HA-eLAR-Myc was immunoprecipitated using an anti-HA antibody, followed by the detection of nidogens (**E**). Non-specific antibodies bound to beads and empty beads were used as controls; 5% input was loaded. (**F–H**) Western blots showing the interaction of PTPRδ with nidogen-1 and -2, in the presence of VSVG-$H_C$T. Nidogen-1 (**F**), nidogen-2 (**G**) and HA-ePTPRδ-Myc (**H**) were immunoprecipitated from lysates of N2a cells, and co-immunoprecipitates were probed using an appropriate antibody (anti-HA for nidogen immunoprecipitations, and anti-nidogen for HA-eLAR-Myc immunoprecipitations). Non-specific antibodies bound to beads and empty beads were used as negative controls; 5% input was loaded.

of the platelet-derived growth factor (PDGF) receptor (Fig. 1B). When nidogen-1 was transiently expressed with this fusion protein (HA-eLAR-Myc) in mouse neuroblastoma-2a (N2a) cells (Appendix Fig. S1B) and immunoprecipitated in the presence of $H_C$T, we found that eLAR was specifically associated with nidogen-1 (Fig. 1C). Similarly, when nidogen-2 was co-expressed with HA-eLAR-Myc (Appendix Fig. S1B) and subjected to immunoprecipitation, it robustly bound to the extracellular domain of LAR, as compared to non-immune mouse IgG or empty bead controls (Fig. 1D). The direct interaction between nidogens and LAR was further confirmed when reverse co-immunoprecipitations were performed. Indeed, when HA-eLAR-Myc was immunoprecipitated, it associated with both nidogen-1 and -2 (Fig. 1E). This fits well with our unpublished results obtained *via* a proximity biotinylation approach in which LAR was identified as a potential interacting partner of $H_C$T in signalling endosomes. In this approach, $H_C$T was fused to a promiscuous biotin ligase (Roux et al, 2012), and allowed to be taken up in signalling endosomes upon incubation with mouse embryonic stem cell-derived motor neurons. This led to biotinylation, and subsequent identification by mass spectrometry, of proteins present within a 10–15 nm radius of the $H_C$T fusion protein in neuronal endosomes, one of which was LAR (SS Novoselov and G Schiavo, unpublished data). Taken together, these results suggest that there is a direct physical interaction between nidogens and LAR, which is preserved upon their entry into signalling endosomes.

Since LAR, PTPRδ and PTPRσ share a high degree of homology and structural similarity (Fig. 1A), we tested whether PTPRδ and PTPRσ also bind to nidogens. As described above, we co-expressed HA-ePTPRδ-Myc and nidogens in N2a cells (Appendix Fig. S1B) and found that the extracellular domain of PTPRδ co-immunoprecipitated with both nidogen-1 and -2 in cell extracts (Fig. 1F,G). Similarly, when HA-ePTPRδ-Myc was immunoprecipitated in the presence of $H_C$T, it was found to bind to both nidogens (Fig. 1H). In contrast, recombinant HA-ePTPRσ-Myc could not be co-immunoprecipitated with either nidogen-1 or -2, suggesting a lack of interaction between nidogens and PTPRσ (Appendix Fig. S1B–D).

## LAR is co-distributed with the nidogen-$H_C$T complex in signalling endosomes of motor neurons

After confirming the direct association between LAR and nidogens, we wanted to assess whether the nidogen-$H_C$T complex and LAR

are internalised and transported together in signalling endosomes. Primary motor neurons were incubated with AlexaFluor 647-labelled $H_C$T ($H_C$T-647) and an antibody against nidogen-2, which were allowed to internalise at 37 °C. After removing the surface-bound probes using a mild acidic wash, the total cellular pool of LAR was detected using an anti-LAR antibody. After immunostaining with appropriate fluorescent secondary antibodies, we observed that the axonal population of LAR displays a punctate pattern reminiscent of endosomal compartments. Crucially, these LAR puncta co-localise with $H_C$T and nidogen-2 in axons and cell bodies, with many of them being triple positive (Fig. 2A,B). This result strongly suggests that all three proteins were co-internalised and underwent long-distance retrograde transport in signalling endosomes. These observations are in line with our previous study that reported the presence of LAR phosphatases in $H_C$T-containing signalling endosomes purified from mouse embryonic stem cell-derived motor neurons (Debaisieux et al, 2016). Upon analysing fluorescence intensities of nidogen-2 and LAR in individual neurites in these cultures, we found that cells with higher levels of LAR contained, on average, higher levels of nidogen-2 in signalling endosomes, as shown by the significant correlation between the levels of internalised nidogens and endogenous LAR (Spearman coefficient = 0.497; Fig. 2C). This correlation was stronger for internalised $H_C$T and LAR, (Spearman coefficient = 0.743; Fig. 2C) indicating that cellular LAR levels directly correlate with endosomal levels of the nidogen-$H_C$T complex.

## Depletion of LAR inhibits $H_C$T entry in motor neurons

If the nidogen-TeNT complex is indeed a ligand of the surface LAR receptor, we reasoned that a decrease in LAR levels would lead to an inhibition of $H_C$T binding and uptake. To test this, we transduced ventral horn cultures with lentiviruses encoding short hairpin RNAs (shRNAs) against mouse LAR. After 48 h of viral transduction, we found that two independent shRNAs were able to reduce endogenous LAR levels by ~70%, compared to empty vector and scrambled controls (Fig. 3A,B). Critically, in this time window, we did not observe any overt alterations in neuronal survival or gross morphology (Fig. 3C). We confirmed the specificity of LAR knockdown by assessing levels of PTPRδ and PTPRσ in these cultures and found that unlike LAR, the levels of these proteins remained unchanged (Figs. 3A and EV1A,B). When these cultures were treated with $H_C$T-647, we found that transduced motor neurons, which were identified by the expression of green

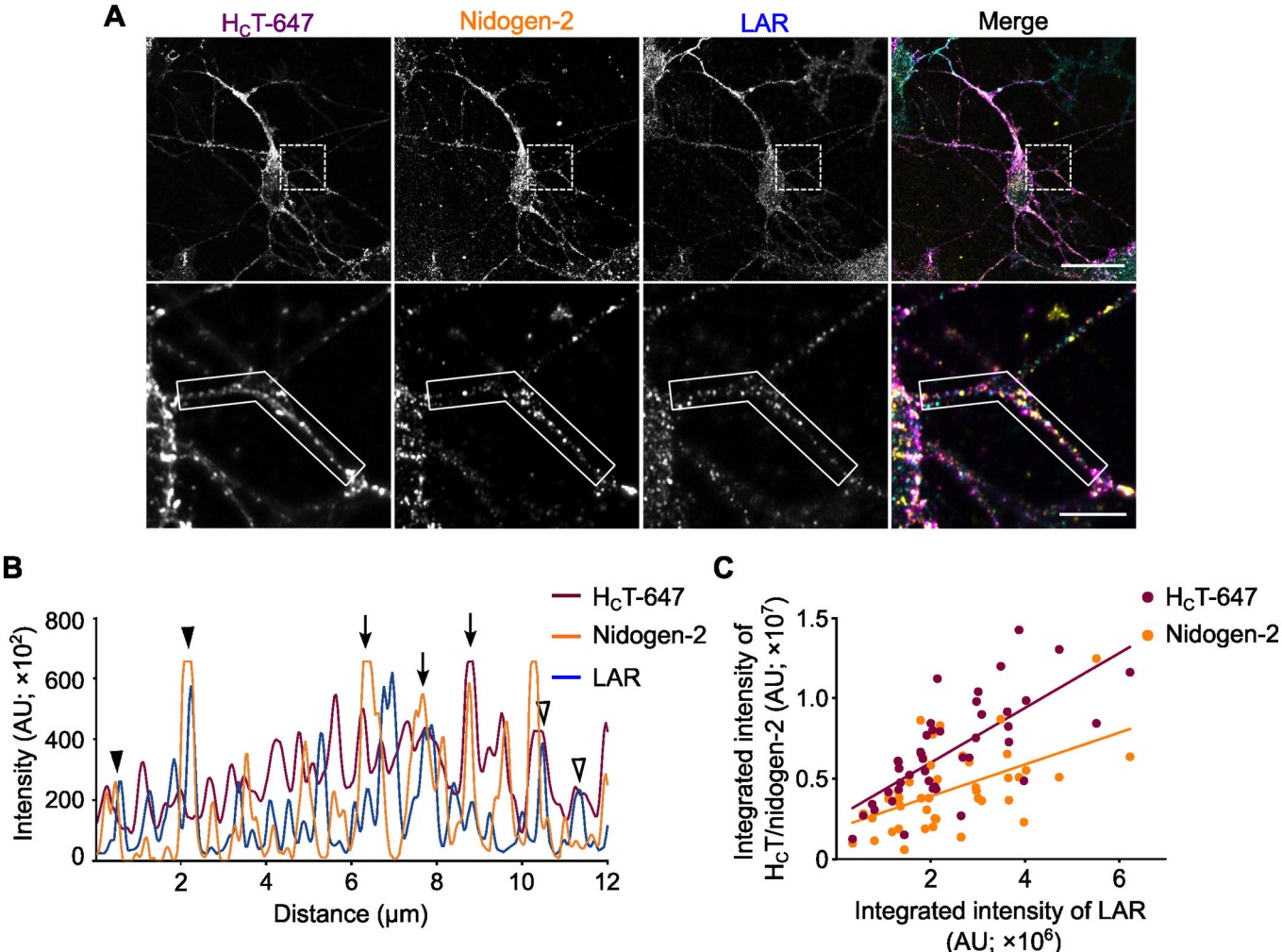

**Figure 2. Endogenous LAR co-localises with the nidogen-H$_C$T complex in signalling endosomes.**

(A) Representative immunofluorescence images of mouse motor neurons treated with H$_C$T-647 and labelled with antibodies against internalised nidogen-2 and total LAR. Images have been pseudo-coloured in magenta (H$_C$T-647), yellow (nidogen-2) and cyan (LAR). A selected region in the upper panel has been magnified in the lower panel. Scale bars: 20 μm (top panel) and 5 μm (bottom panel). (B) Graph showing overlapping intensity profiles of H$_C$T-647, nidogen-2 and LAR in an axonal segment (boxed region in the lower panel of A). Empty arrowheads point to co-localised H$_C$T and LAR organelles, arrowheads denote co-localised nidogen-2 and LAR puncta, while arrows represent puncta containing H$_C$T, nidogen-2 and LAR. (C) Quantification of the neuronal correlation between H$_C$T-647 and nidogen-2 with LAR using fluorescence intensities (*n* = 46 neurites; Spearman coefficient 0.743 and 0.497, and *P* < 0.0001 and 0.0004, for LAR-H$_C$T and LAR-nidogen-2, respectively).

fluorescent protein (GFP), exhibited a significant decrease in H$_C$T internalisation (~40%), compared to controls (Fig. 3C,D). The decrease in H$_C$T uptake was observed using both shRNAs, suggesting that this effect is specific and due to LAR down-regulation rather than potential off-target effects. To confirm this conclusion, motor neurons transduced with LAR shRNA#2 were magnetofected with an HA-eLAR-Myc-expressing plasmid (Fig. 1B). After 24 h, cultures were incubated with H$_C$T-647, fixed, immunostained for GFP, HA and βIII tubulin, and imaged. We found that neurons expressing both shRNA#2 and shRNA-resistant, recombinant eLAR, which were identified by the presence of GFP and HA, respectively, showed a significant increase in endocytosis of H$_C$T, compared to cultures expressing shRNA#2 alone, and was commensurate with H$_C$T levels observed in cultures treated with scrambled shRNAs (Fig. 3C,D). This rescue using the

extracellular LAR domain confirms that the decrease in H$_C$T endocytosis observed upon LAR knockdown is specific and that levels of H$_C$T internalisation directly correlate with neuronal LAR levels.

Tyrosine phosphatases such as LAR are known to regulate the phosphorylation of several tyrosine kinases, ultimately influencing their downstream signalling (Kulas et al, 1996; Sarhan et al, 2016). Accordingly, LAR has previously been shown to interact with the tyrosine kinase TrkB in hippocampal neurons, thus modulating its neurotrophic activity. Neurons devoid of LAR displayed a reduction in the BDNF-induced phosphorylation of TrkB, which led to a decrease in the phosphorylation of its downstream effectors Shc, Akt and ERK, ultimately resulting in increased neuronal death (Yang et al, 2006). Based on this evidence, it was possible that the decrease in H$_C$T uptake observed upon LAR knockdown in motor neurons was indirect and

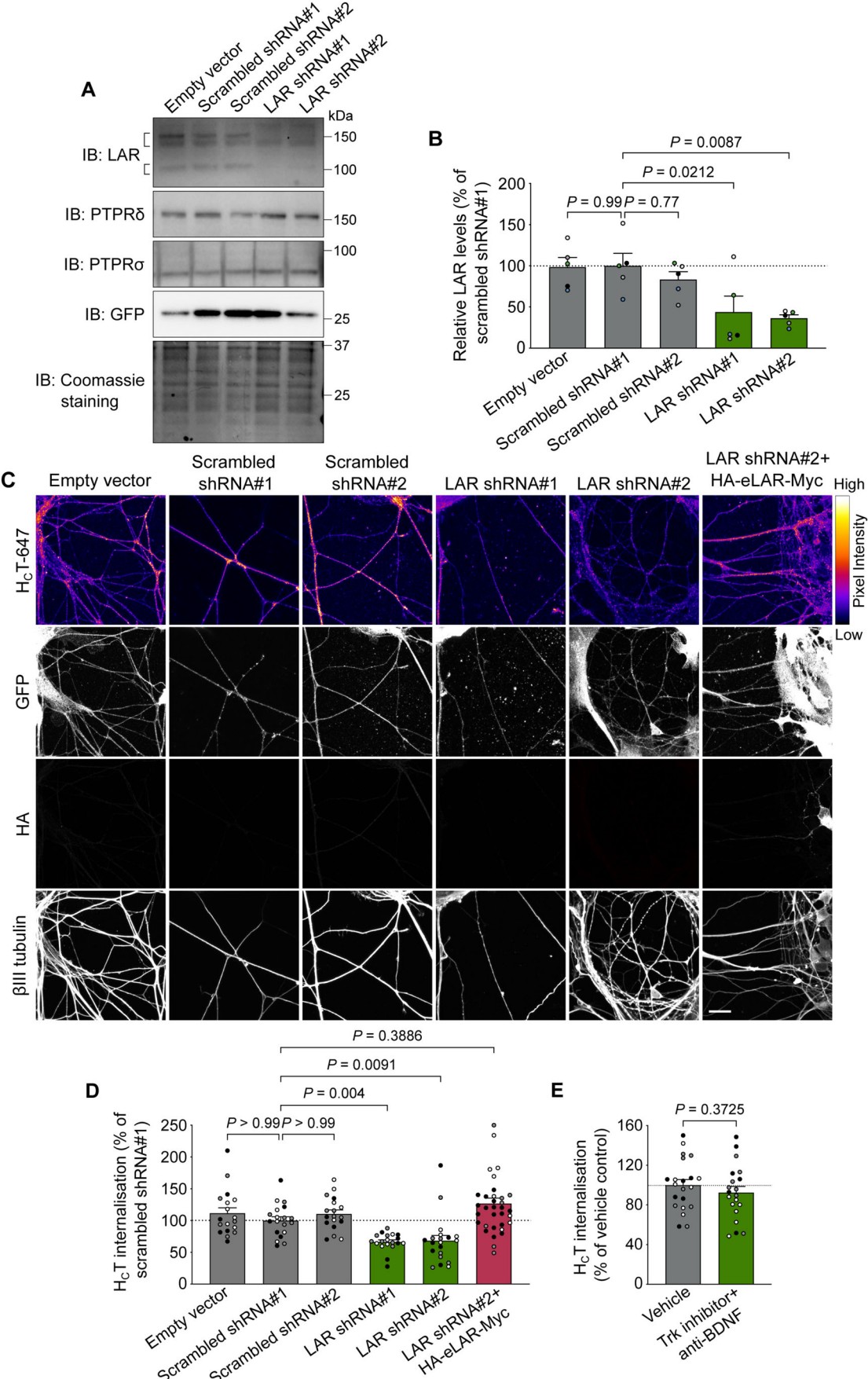

◄ **Figure 3. Depletion of LAR causes a decrease in H$_C$T internalisation in motor neurons.**

(A) Representative western blots for estimating the levels of LAR, PTPRδ and PTPRσ in lysates of ventral horn cultures transduced with lentiviruses encoding short hairpin RNAs (shRNAs) against murine LAR. Lentiviruses carrying an empty vector and two scrambled shRNAs were used as negative controls, whereas GFP was used as a transduction reporter. Coomassie R-250 staining was used to estimate protein loading. (B) LAR quantification shown in (A). Data are presented as a percentage of LAR levels in ventral horn cultures treated with scrambled shRNA#1 ($n = 5$ independent experiments; error bars indicate s.e.m.). Results were tested for statistical significance using one-way analysis of variance (ANOVA; $P = 0.004$), followed by Dunnett's *post-hoc* test. (C) Representative immunofluorescence images showing internalised H$_C$T-647 in motor neurons, following lentiviral-mediated knockdown of endogenous LAR, as well as its rescue by overexpression of shRNA-resistant HA-eLAR-Myc. Images in the H$_C$T-647 channel have been colour mapped based on their intensities. GFP was used as a reporter of lentiviral transduction, while the HA tag was used to confirm expression of HA-eLAR-Myc. Lentiviruses carrying an empty vector and two scrambled shRNAs were used as negative controls. Scale bar: 20 μm. (D) Quantification of endocytosed H$_C$T-647 shown in (C). Data are presented as a percentage of internalised H$_C$T in neurons treated with scrambled shRNA#1 ($n = 3$ independent experiments; error bars indicate s.e.m.). Results were analysed for statistical significance using Kruskal–Wallis test ($P < 0.0001$), followed by Dunn's multiple comparison test. (E) Graph showing levels of internalised H$_C$T in motor neurons treated with the pan-Trk inhibitor PF-06273340 and an anti-BDNF antibody, compared to vehicle control (DMSO). Data are presented as a percentage of internalised H$_C$T in neurons treated with DMSO alone ($n = 3$ independent experiments; error bars indicate s.e.m.). Results were tested for statistical significance using an unpaired *t*-test.

caused by a reduction in TrkB phosphorylation. To rule out this possibility, we tested the effect of PF-06273340, a highly potent and selective inhibitor of Trk receptors (Skerratt et al, 2016). Ventral horn cultures were treated with 100 nM of PF-06273340, together with an anti-BDNF antibody, to abrogate TrkB signalling. HA-H$_C$T was then added, after which the cells were fixed, immunostained for βIII tubulin as well as the HA tag, and imaged. We found that Trk inhibition had no overt effect on H$_C$T internalisation under these experimental conditions (Figs. 3E and EV1C), thus ruling out that the inhibition of nidogen-TeNT uptake observed upon LAR downregulation in motor neurons was mediated by an indirect effect on TrkB signalling.

These results show that the internalisation of H$_C$T requires the expression of LAR in motor neurons and that this role of LAR is independent of its modulation of TrkB. Taken together, these findings suggest that LAR acts as the cellular receptor for the nidogen-H$_C$T complex.

## LAR-nidogen binding is mediated by the fibronectin III domains of LAR

Next, we wanted to characterize the interaction between LAR and nidogen at the molecular level using a protein truncation and co-immunoprecipitation approach. First, the extracellular domain of LAR was truncated into three fragments containing: (i) the Ig domains (HA-LAR Ig1-3-Myc), (ii) the first four FNIII domains (HA-LAR FNIII1-4-Myc) and, (iii) FNIII domains five to eight (HA-LAR FNIII5-8-Myc) (Fig. 4A). Similar to the HA-eLAR-Myc fusion protein, constructs were fused to the transmembrane domain of the PDGF receptor to enable their surface localisation. When these fusion proteins were expressed together with nidogen-2 in N2a cells and immunoprecipitated, we found that nidogen-2 could only be co-immunoprecipitated with constructs containing FNIII domains (Fig. 4B). This result indicated that the LAR-nidogen interaction is mediated exclusively by the FNIII domains of LAR.

We then cloned all eight FNIII domains (HA-LAR FNIIIx-Myc) individually to identify the specific site(s) of interaction with nidogen-2 (Fig. 4A). Co-expression and co-immunoprecipitation analyses demonstrated that the 2$^{nd}$, 4$^{th}$, 5$^{th}$ and 7$^{th}$ FNIII domains efficiently pull down nidogen-2 (Fig. 4C). While the 1$^{st}$ and 8$^{th}$ FNIII domains could also co-precipitate nidogen-2, the immuno-precipitation yield was noticeably lower, suggesting that this interaction is likely to be weaker. In contrast, the 3$^{rd}$ and 6$^{th}$ FNIII domains only displayed non-specific binding of nidogen-2 to the

beads (Fig. 4C). Collectively, these results indicate that the association between LAR and nidogen is likely to be multivalent, i.e., mediated by interactions with multiple FNIII domains across the LAR extracellular domain.

LAR undergoes alternative splicing to generate cell type- and developmental stage-specific isoforms, which display unique interaction profiles. One of these variants, which is specifically expressed in the nervous system, is generated by the alternative splicing of a nine amino acid cassette in the 5$^{th}$ FNIII domain (O'Grady et al, 1994; Zhang and Longo, 1995). The Saito group has previously shown that splicing of this mini-exon, termed MeC, is essential for the in vitro interaction between the 5$^{th}$ FNIII domain and the laminin-nidogen complex (O'Grady et al, 1998). In light of this observation, we wanted to test whether the MeC cassette plays a similar role in the binding of LAR to the nidogen-H$_C$T complex. We deleted this mini-exon from the HA-LAR FNIII5-Myc construct and then assessed the ability of this variant to associate with nidogen-2. We found that nidogen-2 binds both splice variants of the 5$^{th}$ FNIII domain (Fig. 4C), indicating that inclusion of this mini-exon does not alter the ability of LAR to interact with nidogen-2. This observation suggests that the mode of binding of LAR to the nidogen-TeNT complex is different from its interaction with the laminin-nidogen complex and is independent of the MeC mini-exon.

We then wanted to test whether nidogen-1 mirrors the LAR-binding properties of nidogen-2. Co-immunoprecipitations were performed between individual LAR FNIII domains and nidogen-1 under the same experimental conditions described for nidogen-2, which showed that while the overall pattern of binding remained unchanged, there were subtle differences in the strength of the interactions. The 2$^{nd}$, 4$^{th}$, 5$^{th}$ and 7$^{th}$ FNIII domains were the strongest interactors, whereas the 1$^{st}$ FNIII domain displayed a reduced ability to pull down nidogen-1 (Fig. 4D). In contrast, the 8$^{th}$ FNIII domain showed an increased affinity for nidogen-1 than nidogen-2 (Fig. 4D). Akin to nidogen-2, nidogen-1 did not interact with the 3$^{rd}$ and 6$^{th}$ FNIII domains of LAR.

## Recombinant fibronectin III domains of LAR halt the internalisation of the nidogen-H$_C$T complex

Having mapped the interacting domains between LAR and nidogens, we sought to identify the shortest peptide sequences necessary for this interaction, with the aim of using these binding

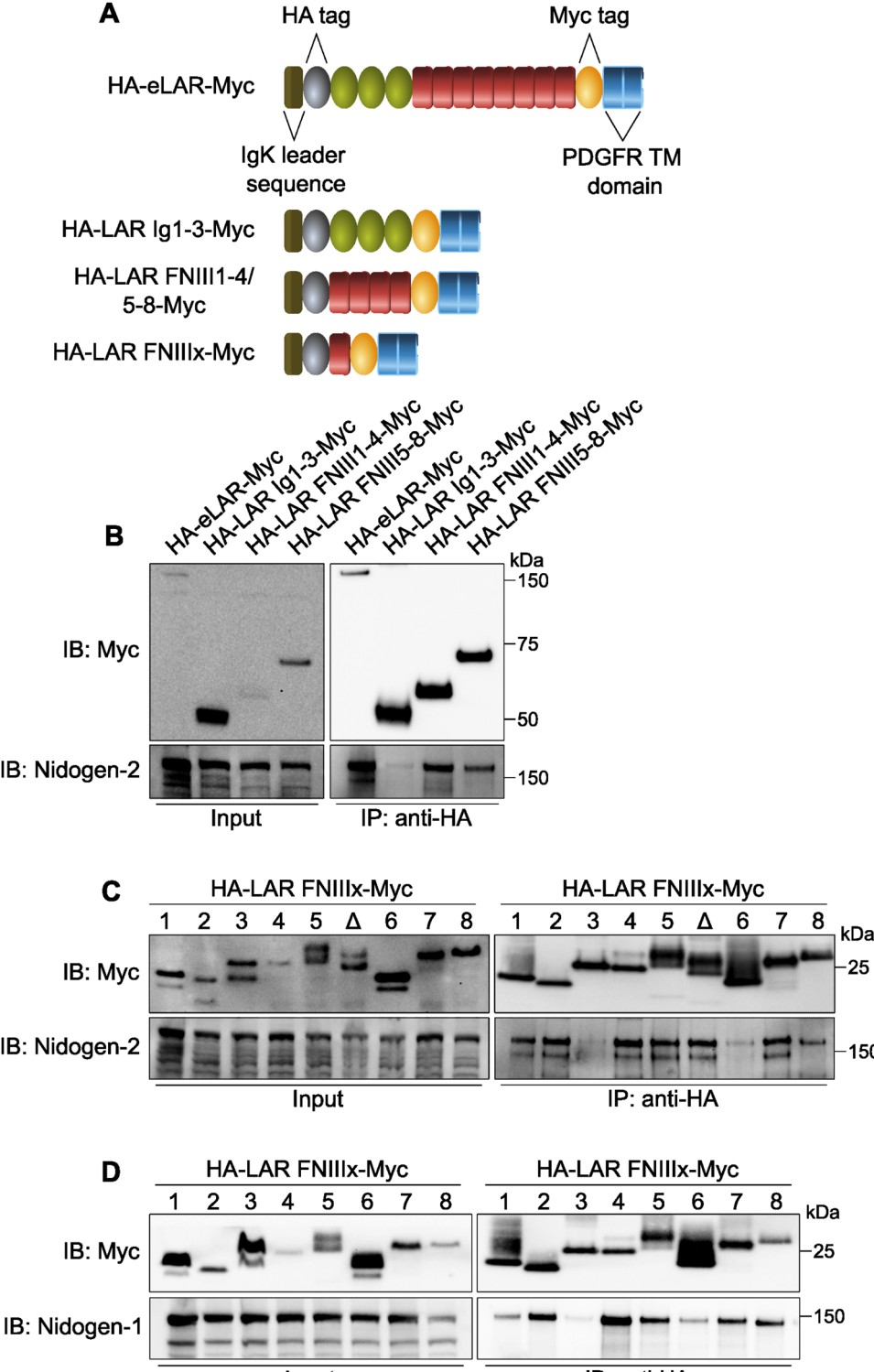

motifs to design competitive inhibitors of the LAR-nidogen interaction. Using the multiple alignment software PRALINE, we aligned the sequences of the 2nd, 4th, 5th and 7th FNIII domains of human LAR and screened for amino acid conservation, as well as for similarities in hydrophobicity/hydrophilicity (Simossis and

Heringa, 2005). Despite possessing a conserved secondary structure, none of these domains were found to contain conserved regions mediating binding to nidogens (Fig. EV2).

In the absence of clear sequence similarities, together with the well-documented modularity and functionality of individual

**Figure 4. Nidogens bind to specific fibronectin III domains of LAR.**

(A) Schematics of LAR fragments used to identify the interacting domains between LAR and nidogens. Truncated proteins were fused to the murine Igκ-chain leader sequence and an HA tag at the N-terminus; the C-terminus was fused to the PDGFR transmembrane domain and a Myc tag. (B) Co-immunoprecipitation and western blot analysis of HA-LAR Ig1-3-Myc, HA-LAR FNIII1-4-Myc and HA-LAR FNIII5-8-Myc with nidogen-2 in the presence of VSVG-$H_C$T. Immunoprecipitation was performed using an anti-HA antibody, and co-immunoprecipitated samples were probed using an anti-nidogen-2 antibody. The HA-eLAR-Myc fusion protein was used as a positive control and 5% input was loaded. (C, D) Western blots showing the interaction between individual LAR FNIII domains and nidogen-2 (C) or nidogen-1 (D), in the presence of VSVG-$H_C$T. The Δ lane refers to co-immunoprecipitations performed using the 5th FNIII domain without the MeC mini-exon. All immunoprecipitations were performed using an anti-HA antibody, while co-immunoprecipitates were probed using an appropriate anti-nidogen antibody. 5% input was loaded.

fibronectin III domains (Petersen et al, 1983; Vilstrup et al, 2020), we decided to use full-length nidogen-binding domains of LAR as competitive inhibitors of the LAR-nidogen interaction. To achieve this, we first cloned the 2nd, 4th, 5th and 7th FNIII domains of LAR into a bacterial expression vector, such that each recombinant protein is fused to a 6×His tag at the N-terminus and a FLAG tag at the C-terminus (Fig. EV3A). Upon bacterial expression, each domain was purified using $Ni^{2+}$-based affinity purification (Appendix Figs. S2, S3) (Vilstrup et al, 2020). We then used enzyme-linked immunosorbent assay (ELISA) to characterise the binding of each of these domains to full-length nidogen-2. Varying concentrations of each FNIII domain were added to 0.5 picomoles of recombinant mouse nidogen-2 in solution, which was captured using an anti-nidogen-2 antibody. Subsequent complex detection showed that purified LAR FNIII domains bound to nidogen-2, indicating that they are functional. Complex formation increased as a function of the in vitro FNIII domain concentration and followed a sigmoidal curve (Fig. EV3). Using this approach, we estimated that the average binding affinity of LAR FNIII domains to nidogen-2 was ~2 μM. Of these, the 5th FNIII domain was the strongest binder, with an apparent binding affinity of ~1.4 μM, while the 7th FNII domain bound with an apparent affinity of ~2.5 μM. The 2nd and 4th FNIII domains each displayed an apparent association constant of ~2 μM and ~1.9 μM, respectively (Fig. EV3).

Next, we tested whether these domains were individually able to bind to endogenous nidogens, potentially blocking the LAR-nidogen interaction and thus acting as competitive inhibitors of the uptake of the nidogen-$H_C$T complex. Since our previous experiments revealed an apparent binding affinity of ~2 μM, we decided to pre-incubate motor neurons with a tenfold excess of each purified FNIII domain (20 μM), after which they were briefly pulsed with $H_C$T-555 and an anti-nidogen-2 antibody. Following media replacement, plasma membrane-bound $H_C$T was allowed to internalise and undergo long-distance transport. Firstly, we found that addition of these recombinant domains did not affect cell health and morphology, as evidenced by βIII tubulin staining (Fig. 5A). Addition of the 2nd, 4th and 7th FNIII domains led to a ~25% decrease in the uptake of nidogen-2 in axons, compared to buffer-treated controls (Fig. 5A,B). However, the 5th FNIII domain, which had the highest binding affinity to nidogen-2 among these domains, did not block nidogen-2 internalisation (Fig. 5B). Importantly, these results were recapitulated upon quantification of $H_C$T-555 uptake in these cells. Similar to nidogen-2, the 2nd, 4th and 7th FNIII domains led to a ~30% decrease in the uptake of $H_C$T, whereas the 5th FNIII domain showed no effect (Fig. 5A,C). These results suggest that whereas individual, recombinant LAR FNIII domains are capable of interacting with nidogens, their efficiency in interfering with the LAR-nidogen interaction in a cellular context is limited.

Following this, we wanted to assess whether soluble FNIII domains added together can bind more efficiently to endogenous nidogens, thus abrogating the LAR-nidogen interaction. Ventral horn cultures were pulsed with $H_C$T-555 and anti-nidogen-2 antibody in the presence of three concentrations of recombinant FNIII domains: 0.25 μM, 10 μM and 20 μM of each domain. After a 45 min chase, cells were fixed and immunostained for βIII tubulin and internalised nidogen-2. Compared to buffer-treated cells, we found a consistent decrease of ~40% in the amount of internalised nidogen-2 in cells incubated with 10 μM and 20 μM of all four recombinant FNIII domains (Fig. 6A,B). This extent of inhibition was higher than that observed using any single domain on its own. Interestingly, this decrease was observed even at the lower dose of each FNIII domain (0.25 μM), suggesting that the LAR-nidogen interaction might be governed by the combined avidity of individual FNIII moieties. When the intensity of internalised $H_C$T was assessed in these cells, we found that its uptake was reduced by ~50% compared to controls (Fig. 6A,C).

Given the enhanced ability of multiple FNIII domains to abolish binding of the nidogen-$H_C$T complex to endogenous LAR, we decided to combine these domains into two recombinant fragments, namely the FNIII1-4-FLAG and FNIII5-7-FLAG fragments (Fig. EV4A). As with the individual FNIII domains, each of these fragments was expressed and purified from bacteria using affinity purification (Appendix Fig. S4), following which ELISA was performed to estimate apparent binding affinities to nidogen-2. We found that while the LAR FNIII1-4-FLAG fragment had an association constant of ~5.6 μM, the FNIII5-7-FLAG fragment bound to recombinant nidogen-2 with an apparent affinity of ~6.8 μM (Fig. EV4B,C). We then tested the ability of these soluble fragments to act as competitive inhibitors of the LAR-nidogen interaction in a cellular context. Since previous experiments had suggested that protein avidity, rather than affinity, plays a primary role in the LAR-nidogen interaction and that FNIII domain concentrations equivalent to one-tenth of their in vitro affinities were sufficient to inhibit this binding, we decided to pulse-chase neurons with $H_C$T-555 and an anti-nidogen-2 antibody in the presence of 0.56 μM of LAR FNIII1-4-FLAG and 0.68 μM of LAR FNIII5-7-FLAG mixed together. As expected, addition of these fragments was equally effective in blocking the internalisation of the nidogen-$H_C$T complex, as shown by a ~35% decrease in nidogen-2 and $H_C$T fluorescence intensity levels in motor neurons (Fig. 6D–F).

Taken together, our data showing the ability of soluble LAR FNIII domains to abolish the binding of the nidogen-$H_C$T complex to endogenous LAR and block its internalisation, strongly indicate that LAR acts as the receptor of the nidogen-TeNT complex in motor neurons.

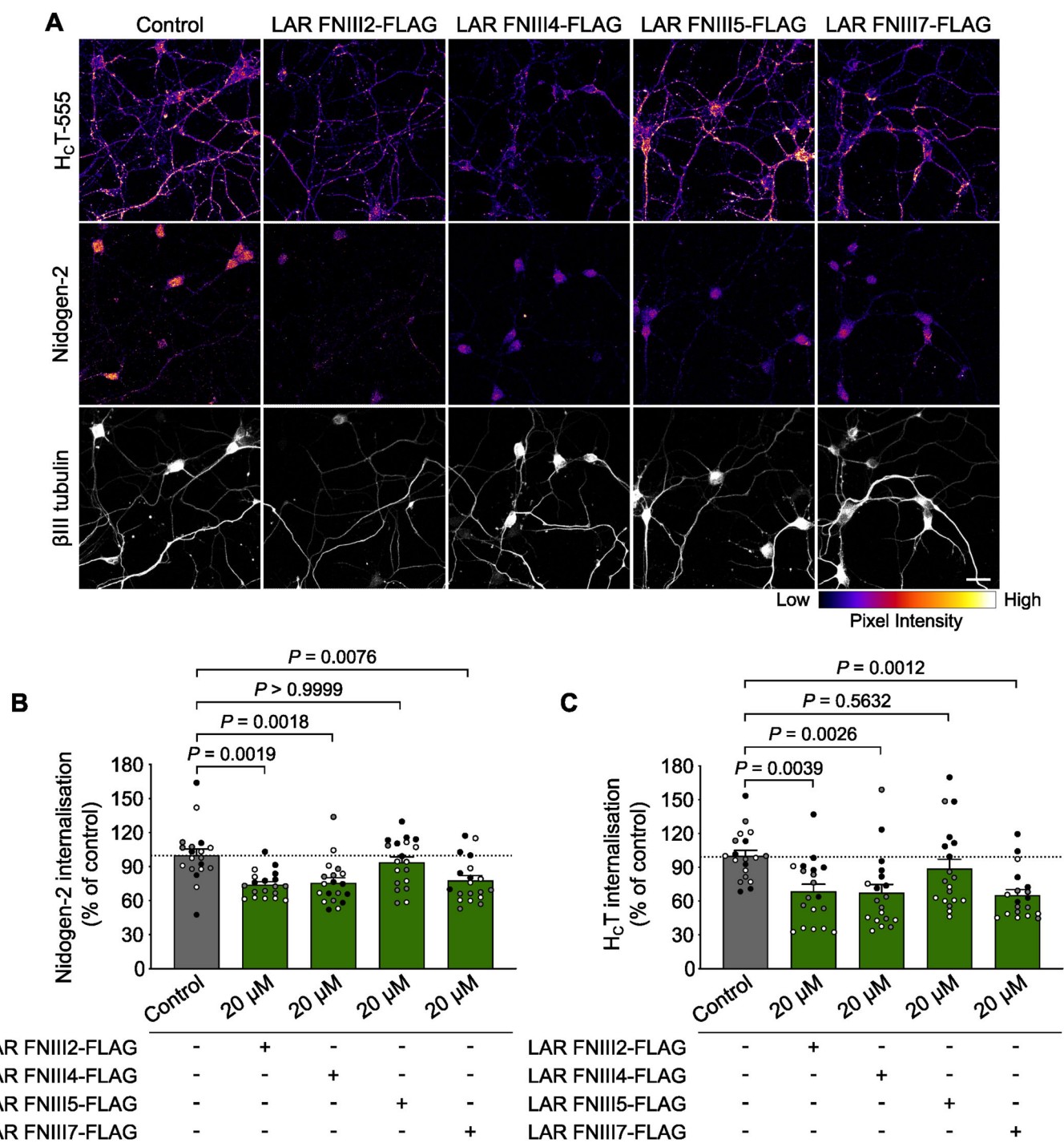

**Figure 5. Individual LAR fibronectin III domains display a limited effect in inhibiting the binding of the nidogen-HcT complex to endogenous LAR.**

(A) Representative immunofluorescence images of motor neurons upon internalisation of $H_CT$-555 and nidogen-2 in the presence of 20 μM recombinant LAR FNIII2, FNIII4, FNIII5 or FNIII7 domains. Images in the top two panels have been colour mapped based on their intensities. Scale bar: 20 μm. (B, C) Graphs showing quantification of endocytosed nidogen-2 (B) and $H_CT$-555 (C) shown in panel (A). Control refers to cultures treated with buffer alone. Data are presented as a percentage of internalised nidogen-2 or $H_CT$ in buffer-treated motor neurons (n = 3 independent experiments; error bars indicate s.e.m.). Results were tested for statistical significance using Kruskal–Wallis test (P = 0.0001), followed by Dunn's *post-hoc* test (B) and one-way ANOVA (P = 0.0005), followed by Dunnett's multiple comparisons test (C).

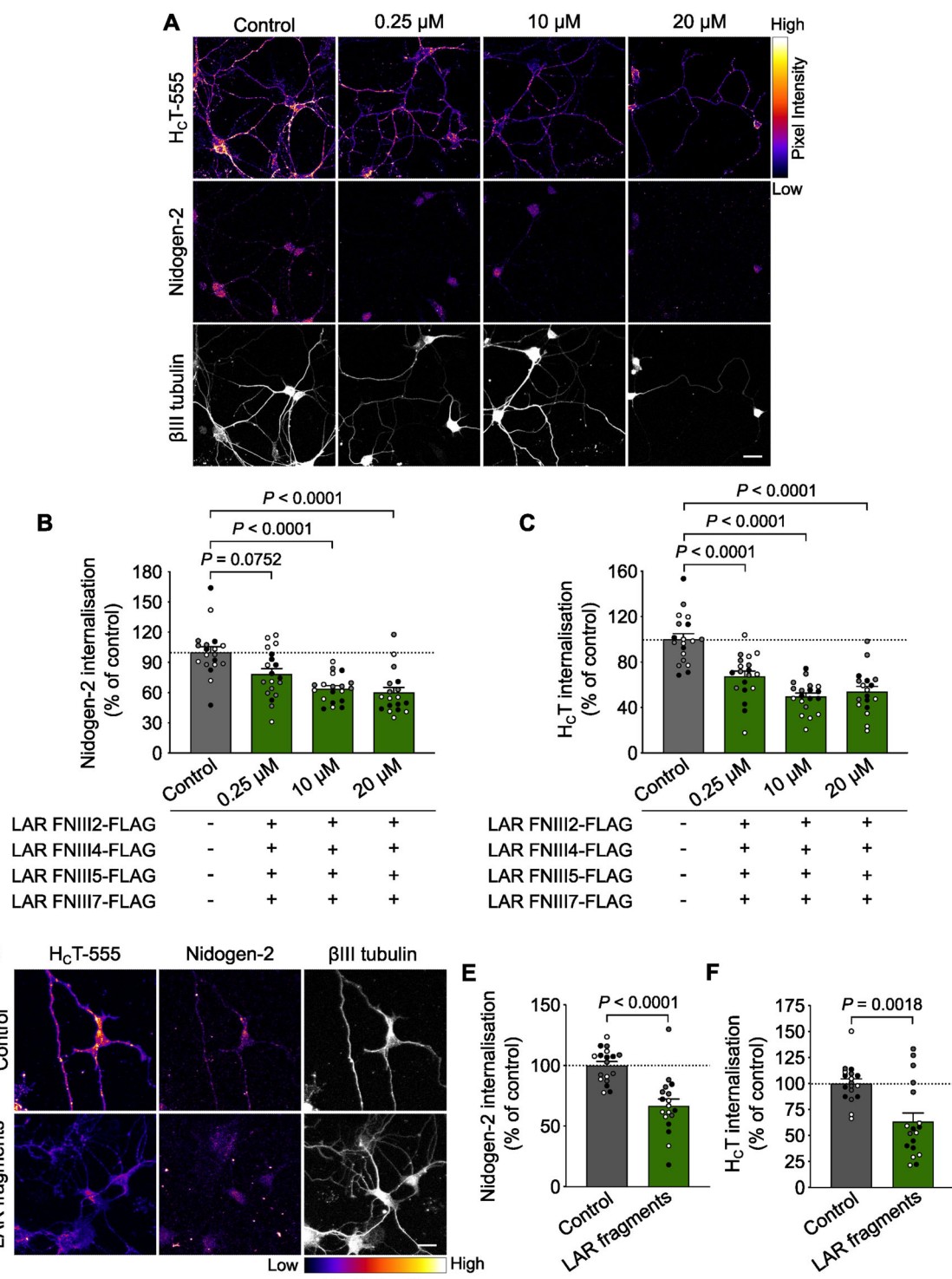

## PTPRδ is a component of the H$_C$T receptor complex in motor neurons

In our previous experiments, we consistently observed that interfering with the LAR-nidogen interaction led to a ~40–50% decrease in H$_C$T internalisation. This indicates that the nidogen-TeNT complex relies on additional membrane proteins for its

endocytosis. The structural and sequence similarities between LAR and PTPRδ, together with our observation that PTPRδ binds to both nidogens in the presence of H$_C$T (Fig. 1F–H), strongly suggested that this protein might be a neuronal receptor of the nidogen-H$_C$T complex and enable its entry into motor neurons.

To first confirm the presence of PTPRδ in H$_C$T-containing signalling endosomes, primary motor neurons were treated with

**Figure 6.  Multiple recombinant LAR fibronectin III domains block nidogen-H$_C$T binding to motor neurons.**

(A) Representative immunofluorescence images of motor neurons upon internalisation of H$_C$T-555 and nidogen-2 in the presence of 0.25 μM, 10 μM and 20 μM of each nidogen-binding FNIII domain (LAR FNIII2, FNIII4, FNIII5 and FNIII7). Images in the top two panels have been colour mapped based on their intensities. Scale bar: 20 μm. (B, C) Quantification of endocytosed nidogen-2 (B) and H$_C$T-555 (C) shown in panel (A). Control refers to cultures treated with buffer alone. Data are presented as a percentage of internalised nidogen-2 or H$_C$T in buffer-treated motor neurons ($n = 3$ independent experiments; error bars indicate s.e.m.). Results were analysed for statistical significance using Kruskal–Wallis test ($P < 0.0001$), followed by Dunn's *post-hoc* test in (B) and one-way ANOVA ($P < 0.0001$), followed by Dunnett's multiple comparisons test in (C). (D) Representative immunofluorescence images of motor neurons upon internalisation of H$_C$T-555 and nidogen-2 in the presence of 0.56 μM of LAR FNIII1-4-FLAG and 0.68 μM of LAR FNIII5-7-FLAG mixed together. Images in the H$_C$T and nidogen-2 panel have been colour mapped based on their intensities. Scale bar: 20 μm. (E, F) Quantification of endocytosed nidogen-2 (E) and H$_C$T-555 (F) shown in panel (D). Control refers to cultures treated with buffer alone. Data are presented as a percentage of internalised nidogen-2 or H$_C$T in buffer-treated motor neurons ($n = 3$ independent experiments; error bars indicate s.e.m.). Results were analysed for statistical significance using an unpaired *t*-test in (E) and Mann–Whitney test in (F).

H$_C$T-555 and an anti-nidogen-2 antibody, and then immunostained for endogenous PTPRδ and internalised nidogen-2. We observed that similar to LAR, PTPRδ exhibited a punctate pattern in neurites (Fig. 7A). These puncta were found to co-localise extensively with both H$_C$T and nidogen-2 (Fig. 7A,B), indicating that H$_C$T, nidogens and PTPRδ share internalisation and endosomal trafficking routes in neurons. Furthermore, we discovered that fluorescence intensities of nidogen-2 and H$_C$T-555 in individual neurites correlate with PTPRδ fluorescence intensities (Spearman coefficient = 0.403 and 0.499, respectively; Fig. 7C), as found previously for LAR.

Next, we wanted to characterise the molecular interaction between nidogens and PTPRδ. The extracellular domain of PTPRδ was divided into three fragments containing: (i) the Ig domains (HA-PTPRδ Ig1-3-Myc), (ii) the first four FNIII domains (HA-PTPRδ FNIII1-4-Myc) and, (iii) FNIII domains five to eight (HA-PTPRδ FNIII5-8-Myc) (Fig. EV4D). When these membrane-anchored fusion proteins were expressed in N2a cells along with nidogen-2 and immunoprecipitated, we found that similar to LAR, nidogen-2 bound to the FNIII5-8-containing PTPRδ fragment. However, unlike LAR, there was no detectable binding to HA-PTPRδ FNIII1-4-Myc. Instead, it was the Ig domain-containing fragment of PTPRδ that was able to immunoprecipitate nidogen-2 (Fig. 7D), indicating that despite the high degree of homology between LAR and PTPRδ, their mode of binding to nidogens is distinct.

To check whether PTPRδ is indeed a receptor of the nidogen-H$_C$T complex, we decided to employ a competitive inhibition assay for PTPRδ, similar to that used for LAR. The PTPRδ Ig1-3 fragment was cloned into a mammalian expression vector such that the resulting recombinant protein is fused to a 6×His tag at the C-terminus (Fig. EV4E); post-expression, this fragment was purified using Ni²⁺-based affinity purification (Appendix Fig. S5A,B). The PTPRδ FNIII5-7 fragment, on the other hand, was purified from bacteria (Fig. EV4F; Appendix Fig. S5C,D), as outlined for LAR domains. An ELISA of PTPRδ Ig1-3-His and FNIII5-7-FLAG with purified nidogen-2 indicated that both recombinant proteins have an apparent association constant of ~2 μM and ~10 μM, respectively (Fig. EV4G,H). When these fragments were added together to motor neurons at concentrations equivalent to one-tenth of their apparent binding affinity (0.2 μM and 1 μM), we found that they had no effect on neuronal health or morphology, as shown by the βIII tubulin staining in these cultures (Fig. 7E). When cultures were treated with H$_C$T-555 and an antibody against nidogen-2 in the presence of these fragments, we found that there was a ~26% decrease in the uptake of nidogen-2 in

axons, compared to buffer-treated controls (Fig. 7E,F). When the intensity of internalised H$_C$T was assessed in these cells, we found that its uptake was reduced by ~40% compared to controls (Fig. 7E,G), thus recapitulating the inhibition in nidogen internalisation. These results suggest that, together with LAR, PTPRδ is a receptor for the nidogen-H$_C$T complex.

## Nidogen-binding fragments derived from LAR and PTPRδ block TeNT-induced paralysis in mice

If LAR and PTPRδ are indeed components of the H$_C$T receptor complex, then their nidogen-binding fragments should inhibit binding and uptake of the nidogen-TeNT complex at the NMJ in vivo. To test this hypothesis, full-length TeNT was injected into the gastrocnemius muscle either alone or in combination with LAR and PTPRδ fragments, and the resulting spastic paralysis was assessed using a footprint assay (Fig. 8A). Gait coordination was monitored by measuring the distance between the injected hind paw and the ipsilateral fore paw (Fig. 8A; black bracket in the control panel) (Bercsenyi et al, 2014), whereas step width was quantified by measuring the distance between the injected and un-injected hind paws (Fig. 8A; red bracket in the control panel) (Moritz et al, 2019). Mice injected with a sub-lethal dose of TeNT developed severe gait abnormalities at 96 h post-injection, with permanent plantar flexion of the affected hind paw (Movies EV1, 2). This was accompanied by hyperextension and flexion of the injected leg during a tail suspension assay and by a significant decrease in gait coordination as well as step width between the hindlimbs in these mice (Figs. 8A–C and EV5). Co-administration of TeNT with the LAR fragments FNIII1-4-FLAG and FNIII5-7-FLAG produced a marked improvement in gait and posture during tail suspension, resulting in the restoration of step width and coordination (Figs. 8A–C and EV5; Movie EV3). Similar observations were made when TeNT was co-injected with the PTPRδ fragments Ig1-3-His and FNIII5-7-FLAG (Figs. 8A–C and EV5; Movie EV4). To test the possibility of an additive effect of LAR and PTPRδ, mice were administered with TeNT and the four nidogen-binding fragments of both tyrosine phosphatases. This group of mice displayed minimal gait abnormalities, with the injected hindlimb extended away from the midline and splayed normally during tail suspension (Fig. EV5; Movie EV5). The coordination distance and step width of these mice were indistinguishable from control mice, indicating a near-complete prevention of TeNT-induced spastic paralysis (Fig. 8A–C).

Finally, we wanted to investigate the development of local tetanic paralysis in the above experimental groups as a function of time. At 24 and 48 h post-injection, TeNT-injected mice displayed

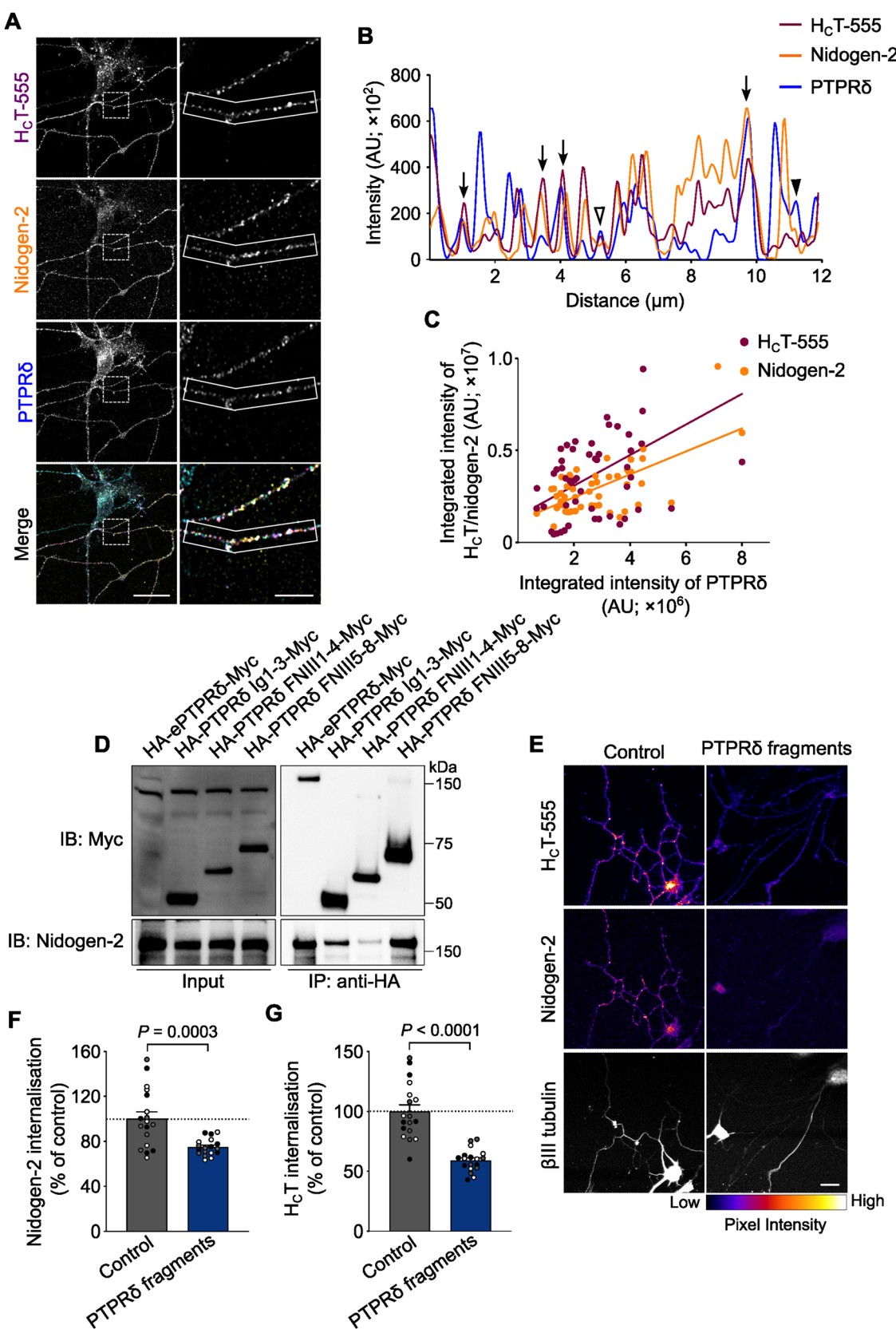

**Figure 7.  Nidogen-binding immunoglobulin and fibronectin III domains of PTPRδ inhibit the internalisation of the nidogen-H$_C$T complex in motor neurons.**

(A) Representative immunofluorescence images of mouse motor neurons treated with H$_C$T-555 and labelled with antibodies against internalised nidogen-2 and total PTPRδ. Images have been pseudo-coloured in magenta (H$_C$T-555), yellow (nidogen-2) and cyan (PTPRδ). Selected region in the left panel has been magnified in the right panel. Scale bars: 20 μm (left panel) and 5 μm (right panel). (B) Graph showing overlapping intensity profiles of H$_C$T-555, nidogen-2 and PTPRδ in an axonal segment (boxed region in the right panel of A). Empty arrowheads point to co-localised H$_C$T and PTPRδ organelles, arrowheads denote co-localised nidogen-2 and PTPRδ puncta, while arrows indicate puncta containing H$_C$T, nidogen-2 and PTPRδ. (C) Quantification of the neuronal correlation between H$_C$T-555 and nidogen-2 with PTPRδ in motor neurons using fluorescence intensities ($n = 51$ neurites; Spearman coefficient 0.4036 and 0.4992, and $P = 0.0033$ and 0.0002, for PTPRδ-H$_C$T and PTPRδ-nidogen-2, respectively). (D) Co-immunoprecipitation and western blot analysis of HA-PTPRδ Ig1-3-Myc, HA-PTPRδ FNIII1-4-Myc and HA-PTPRδ FNIII5-8-Myc with nidogen-2 in the presence of VSVG-H$_C$T. Immunoprecipitation was performed using an anti-HA antibody, and co-immunoprecipitated samples were probed using an anti-nidogen-2 antibody. The HA-ePTPRδ-Myc fusion protein was used as a positive control; 5% input was loaded. (E) Representative immunofluorescence images of motor neurons upon internalisation of H$_C$T-555 and nidogen-2 in the presence of 0.2 μM of PTPRδ Ig1-3-His and 1 μM of PTPRδ FNIII5-7-FLAG mixed together. H$_C$T and nidogen-2 images have been pseudo-coloured based on their intensities. Scale bar: 20 μm. (F, G) Quantification of endocytosed nidogen-2 (F) and H$_C$T-555 (G) shown in panel (E). Control refers to cultures treated with buffer alone. Data are presented as a percentage of internalised nidogen-2 or H$_C$T in buffer-treated motor neurons ($n = 3$ independent experiments; error bars indicate s.e.m.). Results were analysed for statistical significance using an unpaired t-test.

subtle defects in walking. By 72 h, however, there was overt paralysis of the hindlimb, which worsened at 96 h, resulting in a ~65% and ~90% decrease in limb coordination, compared to control mice (Fig. 8D). Due to the permanent flexion of the hindlimb in TeNT-treated mice, there was a ~70% and ~75% decrease in the distance between the hindlimbs at these later timepoints (Fig. 8E). This was accompanied by severe defects in limb splaying and posture. In mice co-injected with TeNT and LAR fragments, the development of spastic paralysis was much slower. Gait and posture abnormalities were less severe at later timepoints, with a negligible change in coordination up to 48 h, which then decreased by ~16% and ~47% at 72 and 96 h, respectively. Step width abnormalities also showed a similar kinetic, with a decrease of ~7% and ~35%, at 72 and 96 h, respectively. Similar changes were observed for the PTPRδ experimental group (Fig. 8D,E). In contrast, mice treated with both LAR and PTPRδ fragments showed a maximal decrease of ~19% and ~10% in coordination and step width at 72 and 96 h, respectively (Fig. 8D,E). Together, these experiments indicate that both LAR and PTPRδ act as receptors of the nidogen-TeNT complex in vivo.

## Discussion

TeNT is a highly potent neurotoxin that causes spastic paralysis by inhibiting neurotransmission in spinal cord inhibitory interneurons. Entry into the central nervous system is achieved by targeting the mammalian NMJ, which leads to its internalisation into motor neurons and subsequent transcytosis into interneurons (Schiavo et al, 2000; Surana et al, 2018), albeit its peripheral action at the NMJ has been also detected (Fabris et al, 2023). Due to the high toxicity of TeNT and its causal role in tetanus disease, a precise understanding of the physiological determinants enabling its entry into the nervous system is urgently needed. Previous studies have shown that binding of TeNT to the NMJ requires the presence of surface polysialogangliosides. However, protein(s) also play an essential role in concentrating TeNT at the NMJ and enabling its entry into motor neurons (Montecucco et al, 2004). Indeed, the ECM proteins nidogens were found to bind TeNT, thereby suggesting a mechanism involving the capture of TeNT at the NMJ and facilitating its neuronal entry (Bercsenyi et al, 2014). However, the identity of the membrane receptor that binds to the nidogen-TeNT complex on the surface of motor neurons and

targets it to long-distance axonal transport is currently unknown. In this study, we show that the transmembrane tyrosine phosphatases LAR and PTPRδ directly interact with the nidogen-TeNT complex and ferry it into motor neurons, thus acting as its neuronal receptors.

LAR, PTPRδ and PTPRσ have been described as synaptic organisers that play important roles in the developing and mature nervous system, including axon guidance and neurite extension, as well as synapse formation, differentiation, and plasticity (Cornejo et al, 2021). The receptor-like extracellular domain has been reported to interact with a variety of trans-synaptic ligands and ECM molecules, thereby modulating cell adhesion. These include netrin-G ligand-3, heparan sulphate proteoglycans and laminin in the case of LAR (O'Grady et al, 1998; Johnson and Van Vactor, 2003; Woo et al, 2009), and interleukin-1 receptor accessory protein, interleukin-1-receptor accessory protein-like 1 and Slitrk3 for PTPRδ (Yoshida et al, 2011, 2012; Takahashi et al, 2012). On the other hand, the catalytic phosphatase subunit regulates the phosphorylation status and activity of several proteins, including signalling molecules, such as liprins (Um and Ko, 2013). Therefore, LAR-RPTPs are ideally positioned to act as molecular linkers that couple ECM components with downstream signalling cascades in the nervous system.

The starting point of our study was the previously reported interaction between LAR and nidogens. While Ackley and co-workers had shown a genetic interaction between these proteins, the Saito laboratory demonstrated that the laminin-nidogen complex acts as a ligand for LAR (O'Grady et al, 1998; Ackley et al, 2005). However, neither study showed a direct interaction between these molecules. Here, we have demonstrated that LAR binds to both nidogen-1 and -2 in the presence of H$_C$T. These experiments, which were performed using only the extracellular portion of LAR, suggest that the intracellular phosphatase domain is dispensable for this interaction. This is in line with the observation that several RPTPs carry out their extracellular functions independently of their phosphatase domains (Young et al, 2021). It is, however, currently unclear whether nidogen binding modulates the phosphatase activity of LAR or changes the specificity of its downstream targets. While the LAR homologue PTPRδ also binds to nidogens, PTPRσ does not, confirming previous reports that these proteins are highly selective in their binding properties despite their sequence and structural similarities (Coles et al, 2015).

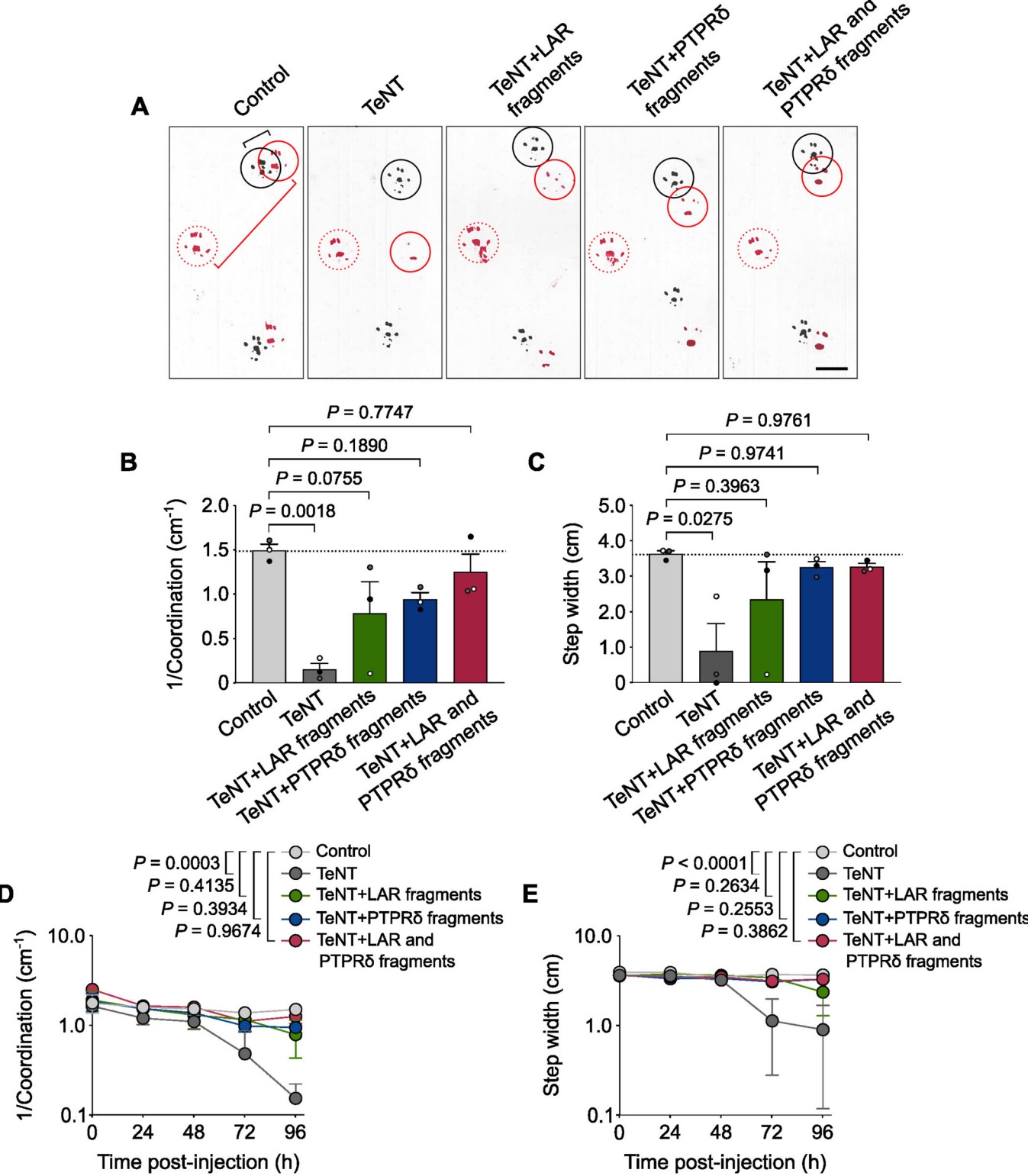

Clostridial neurotoxins, such as TeNT, are known to hijack endogenous trafficking pathways to gain access to the nervous system and evade intracellular degradation (Surana et al, 2018). We have previously reported that, after binding to nidogen-rich regions at the NMJ, TeNT accomplishes its journey from the NMJ to the

spinal cord by hitchhiking on signalling endosomes. These endocytic organelles contain several ligand-receptor complexes, including the neurotrophin BDNF and its receptors TrkB and p75[NTR] (Lalli and Schiavo, 2002; Deinhardt et al, 2006). During its internalisation and intracellular transport, TeNT triggers Trk

◄ **Figure 8. Soluble LAR and PTPRδ fragments inhibit tetanic paralysis in mice.**

(A) Representative footprint tracks 96 h after injection of mice with TeNT pre-incubated with control buffer or with recombinant, nidogen-binding LAR and/or PTPRδ fragments. Mice were injected in the gastrocnemius muscle of the right hindlimb. The footprint of the injected right hindlimb is circled in red, while the footprint of the contralateral, non-injected hindlimb is marked by dotted red circles; black circles show the footprint of the ipsilateral front paw. The black bracket in the control panel denotes the coordination length, while the red bracket shows the step width between hind paws. Scale bar: 1 cm. (B, C) Graphs showing the measured coordination (B) and step width (C) 96 h after injection ($n = 3$ independent experiments; error bars indicate s.e.m.). Results were analysed for statistical significance using one-way ANOVA ($P = 0.0053$ and 0.0507 for (B) and (C), respectively), followed by Dunnett's multiple comparisons test. (D) Quantification of coordination at the indicated timepoints post-injection ($n = 3$ independent experiments; error bars indicate s.e.m.). Results were analysed for statistical significance using two-way ANOVA ($P < 0.0001$ and 0.0001 for treatment and time variables, respectively), followed by Dunnett's multiple comparisons test. (E) Quantification of step width between the hind paws at the indicated timepoints post-injection ($n = 3$ independent experiments; error bars indicate s.e.m.). Results were analysed for statistical significance using two-way ANOVA ($P < 0.0001$ for both treatment and time variables), followed by Dunnett's multiple comparisons test.

receptor phosphorylation, leading to initiation of downstream signalling cascades via activation of phospholipase Cγ-1 and phosphatidylinositol 3-kinase (Gil et al, 2003; Calvo et al, 2012). Concomitantly, multiple studies have reported the localisation of LAR-RPTPs at synaptic regions, with LAR and PTPRδ particularly enriched at excitatory synapses; LAR has also been shown to be abundant at NMJs (Kaufmann et al, 2002; Ackley et al, 2005; Dunah et al, 2005; Park et al, 2020). Interestingly, synaptic accumulation of LAR is dependent on the presence of nidogens in the neuronal basement membrane (Ackley et al, 2005). LAR regulates vesicular trafficking by recruiting synaptic vesicles to active zones and coupling exo-endocytosis (Takahashi and Craig, 2013). All three LAR-RPTPs are present in $H_CT$-containing signalling endosomes in motor neurons (Debaisieux et al, 2016), with both LAR and PTPRδ regulating the BDNF-dependent phosphorylation and activation of TrkB (Yang et al, 2006; Tomita et al, 2020). Our experiments have shown that LAR and PTPRδ show a punctate pattern in motor neurons and are co-distributed with $H_CT$ and nidogen-2, indicating their presence in shared axonal carriers. In keeping with a putative role of LAR as a receptor for the nidogen-TeNT complex, LAR knockdown was sufficient to inhibit $H_CT$ internalisation in motor neurons. This result was confirmed with two independent shRNAs and rescued by expression of an shRNA-resistant extracellular LAR subunit, strongly indicating that this effect was specific and not caused by off-target effects. This receptor function of LAR is independent of its role in regulating the phosphorylation of the TrkB receptor, since incubation with a validated TrkB inhibitor had no effect on $H_CT$ uptake in neurons under our experimental conditions. We also observed that $H_CT$ and nidogen-2 uptake was strongly correlated with the expression levels of LAR and PTPRδ, further lending support to our hypothesis.

To better understand the LAR-nidogen interaction, we truncated the LAR extracellular domain and performed co-immunoprecipitations of these fragments with full-length nidogens. We found that rather than binding to a single site, nidogens make multiple contacts along the length of the LAR extracellular domain. All the identified areas lie on FNIII domains, with the strongest binding shown by the 2nd, 4th, 5th and 7th FNIII domains, pointing to a multivalent interaction. However, we are unable to posit whether this binding is cooperative, that is, whether binding of one FNIII domain to nidogen facilitates binding to additional sites. Similar co-immunoprecipitation experiments with PTPRδ showed that the Ig-rich region, along with the 5th–8th FNIII domains, was responsible for nidogen binding. This confirms previous observations that, despite their sequence and structural

homology, LAR-RPTPs have unique modes of binding to their interacting partners.

The stoichiometry of the RPTP-nidogen interaction also remains unclear. Despite displaying the same structural fold, we were unable to uncover any significant sequence similarity between the LAR 2nd, 4th, 5th and 7th FNIII domains, barring amino acids that are necessary for fibronectin III domain folding. This is in stark contrast to the highly conserved Arg-Gly-Asp tripeptide in the 10th FNIII domain of the ECM protein fibronectin, which is essential for its interaction with integrins in vivo (Takahashi et al, 2007). Nonetheless, the relative in vitro binding affinities of recombinant nidogen-2 and LAR FNIII domains yielded an average value of ~2 μM. This is consistent with the in vitro binding affinities of other surface protein-ECM pairs, which have been reported to rely on multivalent interactions rather than the affinity of single binding events (Wright, 2009). Moreover, these experiments were performed using soluble LAR FNIII domains untethered to a membrane; this contrasts with the interaction taking place in cells where anchoring of FNIII domains on the plasma membrane would result in increased local concentrations of receptor molecules. Interestingly, LAR is sequestered in caveolin-containing membrane microdomains, which are known to be enriched in cholesterol, glyco-sphingolipids and sphingomyelin (Caselli et al, 2002). Interactions with their extracellular binding partners trigger local clustering of LAR-RPTPs on the membrane, leading to formation of higher-order complexes (Um et al, 2014; Won et al, 2017; Xie et al, 2020; Coles et al, 2011). This molecular clustering is similarly observed for polysialogangliosides in lipid microdomains, which act as primary receptors as well as concentrating platforms for TeNT (Prinetti et al, 2000; Herreros et al, 2001). LAR and PTPRδ are thus ideally positioned to bind to TeNT, along with polysialogangliosides and nidogens, and initiate its internalisation and delivery to the central nervous system.

The validity of our hypothesis was further confirmed by the inhibition of $H_CT$ uptake by pre-incubation of motor neuron cultures with soluble LAR FNIII domains. If LAR is indeed a surface receptor for the nidogen-TeNT complex, we reasoned that soluble FNIII domains, when added in excess, would outcompete endogenous LAR and disrupt the LAR-nidogen interaction, leading to a decrease in endocytosis of nidogens as well as $H_CT$. Whereas addition of a single recombinant FNIII domain had limited effect on $H_CT$ and nidogen internalisation, simultaneous addition of multiple FNIII domains proved to be more effective, underscoring that the LAR-nidogen interaction is likely to be driven by the overall avidity of FNIII domains to nidogens. This

conclusion was further supported by the inhibition of nidogen-$H_C$T internalisation using multivalent fragments from LAR and PTPRδ.

Finally, we tested the ability of these LAR and PTPRδ fragments to prevent the spastic paralysis induced in mice injected with TeNT. TeNT-injected mice displayed mild gait and posture abnormalities within 24 h, which drastically worsened 72 h post-injection. However, this was not the case for mice co-injected with TeNT and LAR and/or PTPRδ fragments. These mice displayed milder defects, which worsened gradually over the course of the experiment. Mice injected with all four nidogen-binding fragments showed the least severe phenotypes, compared to LAR or PTPRδ fragments co-injected with TeNT on their own, suggesting an additive effect of LAR and PTPRδ in preventing TeNT intoxication. The near-complete abrogation of spastic paralysis in vivo demonstrates that LAR and PTPRδ form receptor complexes with nidogen-TeNT and play an essential role in ferrying this complex into the nervous system.

Taken together, our results provide the first identification of membrane receptors for TeNT at the NMJ. By identifying LAR and PTPRδ as binding partners of nidogens, we have revealed a neuronal specific pathway by which TeNT is targeted to axonal signalling endosomes and undergoes long-distance transport to the neuronal cell body. The RPTP-nidogen complex, along with polysialogangliosides, thus acts as an efficient capture mechanism for TeNT at the NMJ, which enables its binding at very low concentrations and efficient uptake into motor neurons. While ECM proteins have been reported to bind to a variety of surface molecules, this is the first report of membrane-bound receptors enabling the internalisation and trafficking of ECM proteins in neurons, the physiological relevance of which remains to be uncovered. We have also established a molecular link between the ECM and trophic pathways in the nervous system, suggesting that nidogens might play critical roles in controlling growth factor availability at synapses as well as regulating neuronal signalling. This paves the way for dissecting the mechanisms controlling the uptake of physiological ligands and toxins in neurons, and their targeting to long-distance axonal transport within the nervous system. Importantly, the discovery of these receptor complexes, as well as the identification of specific competitive inhibitors, makes them attractive targets for the development of therapeutics against tetanus.

## Methods

### Key resources

| Reagents | | | |
|---|---|---|---|
| Name | Source | Identifier | |
| Anti-HA magnetic beads | Thermo Fisher Scientific | 88837 | |
| AlexaFluor 555 C2 maleimide | Thermo Fisher Scientific | A20346 | |
| AlexaFluor 647 C2 maleimide | Thermo Fisher Scientific | A20347 | |

| Reagents | | | |
|---|---|---|---|
| Name | Source | Identifier | |
| Brain-derived neurotrophic factor (BDNF) | Peprotech | 450-02-100 | |
| Ciliary neurotrophic factor (CNTF) | Peprotech | 450-50-50 | |
| Coomassie brilliant blue R-250 | Bio-Rad Laboratories | 1610400 | |
| Dulbecco's modified Eagle medium (DMEM) | Gibco | 41966-029 | |
| Expi293 Expression media | Thermo Fisher Scientific | A1435101 | |
| Foetal bovine serum (FBS) | Thermo Fisher Scientific | 10309433 | |
| Gibson assembly cloning kit | New England Biolabs | E5510S | |
| Glial cell line-derived neurotrophic factor (GDNF) | Peprotech | 450-10-50 | |
| GlutaMAX | Thermo Fisher Scientific | 35050061 | |
| iMatrix-511 | Reprocell | NP892-011 | |
| Laminin | Merck | L2020 | |
| Lenti-X concentrator | Takara Bio | 631232 | |
| Lipofectamine 3000 | Invitrogen | L3000008 | |
| Neurobasal medium | Thermo Fisher Scientific | 21103049 | |
| NeuroMag transfection reagent | OZ Biosciences | NM50200 | |
| Opti-MEM reduced media | Thermo Fisher Scientific | 31985062 | |
| Penicillin-streptomycin | Thermo Fisher Scientific | 15-140-122 | |
| PF-06273340 | Merck | PZ0254 | |
| Poly-L-ornithine | Merck | P4957 | |
| Protein A sepharose(R) 4B, fast flow | Merck | P9424 | |
| Protein G sepharose 4, fast flow | Merck | GE17-0618-01 | |

| Reagents | | | |
|---|---|---|---|
| **Name** | **Source** | **Identifier** | |
| Recombinant mouse nidogen-2 protein | R&D Systems | 6760-ND-050 | |
| Retinoic acid | Merck | R2625-50 | |
| 1-Step ultra TMB solution | Thermo Fisher Scientific | 34029 | |

| **Primary and secondary antibodies** | | | |
|---|---|---|---|
| **Name** | **Application** | **Source** | **Identifier** |
| Chicken polyclonal anti-BDNF | Blocking | R&D Systems | AF248; AB_355275 |
| Chicken polyclonal anti-GFP | IF | Aves Labs | GFP-1010; AB_2307313 |
| Chicken polyclonal anti-βIII Tubulin | IF | Synaptic Systems | 302306; AB_2620048 |
| Goat polyclonal anti-ChAT | IF | Merck | AB144P; AB_2079751 |
| Mouse monoclonal anti-FLAG (M2) | ELISA | Merck | F3165; AB_259529 |
| Mouse monoclonal anti-FLAG (FG4R) | IB | Thermo Fisher Scientific | MA1-91878; AB_1957945 |
| Mouse monoclonal anti-GFP (B-2) | IB | Santa Cruz Biotechnology | sc-9996; AB_627695 |
| Mouse monoclonal anti-HA (12CA5) | IB, IP | Cancer Research UK London | |
| Mouse monoclonal anti-6×His (HIS.H8) | ELISA | Abcam | ab18184; AB_444306 |
| Mouse monoclonal anti-LAR | IF | NeuroMabs | 75-193; AB_10675291 |
| Mouse polyclonal anti-LAR (7/LAR) | IB | BD Biosciences | 610350; AB_397740 |
| Mouse monoclonal anti-Myc (9E10) | IB | Thermo Fisher Scientific | 132500; AB_2533008 |
| Mouse polyclonal anti-PTPRσ | IB, IF | MediMabs | MM-0020-P |
| Mouse monoclonal anti-βIII tubulin | IB, IF | BioLegend | 801201; AB_2313773 |
| Mouse IgG | IP | Merck | 12-371; AB_145840 |
| Rabbit polyclonal anti-nidogen-1 | IB, IP | Abcam | ab14511; AB_301290 |
| Rabbit polyclonal anti-nidogen-2 | IB, IF | Abcam | ab14513, AB_301292 |

| Reagents | | | |
|---|---|---|---|
| **Name** | **Source** | **Identifier** | |
| Rabbit polyclonal anti-PTPRδ | IB, IF | Abcam | ab103013; AB_10710803 |
| Rabbit polyclonal anti-βIII tubulin | IF | Merck | T2200; AB_262133 |
| Rabbit IgG | IP | Merck | 12-370; AB_145841 |
| Rat monoclonal anti-HA (3F10) | IF | Roche | 12158167001; AB_390915 |
| Rat monoclonal anti-PTPRδ (F34a6) | IF | Merck | MABS2189 |
| Sheep polyclonal anti-nidogen-2 | ELISA, IP | Schiavo Laboratory | |
| DyLight 405-conjugated donkey anti-chicken IgY | IF | Jackson ImmunoResearch | 703-475-155; AB_2340373 |
| DyLight 405-conjugated donkey anti-mouse IgG | IF | Jackson ImmunoResearch | 715-475-150; AB_2340839 |
| AlexaFluor 405-conjugated goat anti-rabbit IgG | IF | Thermo Fisher Scientific | A31556; AB_221605 |
| AlexaFluor 488-conjugated donkey anti-chicken IgY | IF | Jackson ImmunoResearch | 703-545-155; AB_2340375 |
| AlexaFluor 488-conjugated donkey anti-goat IgG | IF | Thermo Fisher Scientific | A-11055; AB_2534102 |
| AlexaFluor 488-conjugated donkey anti-rabbit IgG | IF | Thermo Fisher Scientific | A21206; AB_2535792 |
| AlexaFluor 488-conjugated donkey anti-rat IgG | IF | Thermo Fisher Scientific | A21208; AB_141709 |
| AlexaFluor 555-conjugated donkey anti-goat IgG | IF | Thermo Fisher Scientific | A-21432; AB_141788 |
| AlexaFluor 555-conjugated donkey anti-mouse IgG | IF | Thermo Fisher Scientific | A31570; AB_2536180 |
| AlexaFluor 555-conjugated goat anti-rat IgG | IF | Thermo Fisher Scientific | A21434; AB_2535855 |
| AlexaFluor 647-conjugated donkey anti-rabbit IgG | IF | Thermo Fisher Scientific | A31573; AB_2536183 |
| AlexaFluor 647-conjugated goat anti-rat IgG | IF | Thermo Fisher Scientific | A-21247; AB_141778 |

| Reagents | | | |
|---|---|---|---|
| **Name** | **Source** | **Identifier** | |
| HRP-conjugated goat anti-mouse IgG | ELISA, IB | Agilent Dako | P0447; AB_2617137 |
| HRP-conjugated goat anti-rabbit IgG | IB | Bio-Rad Laboratories | 1706515; AB_11125142 |
| HRP-conjugated goat anti-mouse IgG (light-chain specific) | IB | Jackson ImmunoResearch | 115-035-174; AB_2338512 |
| HRP-conjugated rat anti-mouse IgG (TrueBlot) | IB | Rockland Immunochemicals | 18-8817-33; AB_2610851 |
| HRP-conjugated mouse anti-rabbit IgG (light-chain specific) | IB | Jackson ImmunoResearch | 211-032-171; AB_2339149 |
| HRP-conjugated mouse anti-rabbit IgG (TrueBlot) | IB | Rockland Immunochemicals | 18-8816-33; AB_2610848 |
| **Nucleic acid sequences** | | | |
| **Sequence** | **Source** | **Identifier** | |
| 5′-gcttcgcgccgtagtctta-3′ | GeneCopoeia | Scrambled shRNA#1 | |
| 5′-tggctgcatgctatgttga-3′ | GeneCopoeia | Scrambled shRNA#2 | |
| 5′-ggatatcgcgtctactatacc-3′ | GeneCopoeia | LAR shRNA#1 | |
| 5′-cctatgaccattctcgagtcc-3′ | GeneCopoeia | LAR shRNA#2 | |

*ELISA* enzyme-linked immunosorbent assay, *IB* immunoblotting, *IF* immunofluorescence, *IP* immunoprecipitation.

## Plasmids and cloning

pDisplay plasmids encoding HA-eLAR-Myc, HA-ePTPRδ-Myc and HA-ePTPRσ-Myc were kindly provided by the Sala laboratory (University of Milan, Italy) (Valnegri et al, 2011). In these plasmids, the extracellular domain of each human RPTP is fused to the murine Igκ chain leader sequence and an HA tag (YPYDVPDYA) at the N-terminus; the C-terminus is fused to a Myc tag (EQKLISEEDL) and the platelet-derived growth factor (PDGF) receptor transmembrane domain, enabling surface localisation of the expressed proteins. pCEP.Pu plasmids encoding mouse nidogen-1 and -2 were provided by Dr. Takako Sasaki (Oita University, Japan) (Bechtel et al, 2012). For this study, nidogen-2 was cloned into a pcDNA3.1 vector and tagged with a 6×His tag at the C-terminus. Plasmids containing LAR and PTPRδ truncations were made using the pDisplay-HA-eLAR-Myc and pDisplay-HA-ePTPRδ-Myc plasmids, respectively. For bacterial expression and purification, FNIII domains/fragments of LAR and PTPRδ were cloned into the pET28a(+) vector (Novagen, 69864), containing an N-terminal 6×His tag and a C-terminal FLAG tag (DYKDDDDK). For mammalian expression and purification, the PTPRδ Ig1-3 fragment was cloned into the pHL-sec vector, containing a C-terminal 6×His tag (Addgene plasmid no. 99845; RRID: Addgene_99845) (Aricescu et al, 2006). Cloning was performed

using Gibson or inverse PCR cloning strategies. Plasmids encoding LAR shRNA and scrambled controls were obtained commercially (GeneCopoeia, MSH030014). Packaging (pCMVR8.74, Addgene plasmid no. 22036, RRID: Addgene_22036) and envelope plasmids (pMD2.G, Addgene plasmid no. 12259, RRID: Addgene_12259) were originally prepared by the Didier Trono laboratory (École Polytechnique Fédérale de Lausanne, Switzerland).

## Cell lines

N2a cells were sourced from Cancer Research UK London Research Institute Cell Services, while Lenti-X HEK293T cells were acquired from ClonTech (632180). Both cell lines were cultured in DMEM with 10% fetal bovine serum (FBS) and 1% GlutaMAX. Cells were split every 2–3 days at 80–90% confluency. For immunoprecipitation experiments, N2a cells cultured for 24 h, and then differentiated using 10 μM retinoic acid for 48–72 h. For immunofluorescence experiments, they were plated on poly-D-lysine-coated coverslips and cultured for 48 h. Lenti-X HEK293T cells were plated directly on Petri dishes for lentiviral production. Expi293F cells were purchased from Thermo Fisher Scientific (A14527) and cultured in Expi293 Expression media.

## Motor neuron cultures

Mixed embryonic ventral horn cultures, referred to in this study as primary motor neurons, were isolated from E11.5–13.5 mouse embryos as previously described (Fellows et al, 2020). Briefly, ventral horns from E11.5–13.5 pregnant wild-type mice (C57Bl6/SJ6, Charles River) were dissociated, centrifuged at 380×g for 5 min, seeded on poly-L-ornithine- and laminin-coated coverslips or wells, and maintained in motor neuron media (Neurobasal with 2% v/v B27, 2% heat-inactivated horse serum, 1× GlutaMAX, 24.8 μM β-mercaptoethanol, 10 ng/ml CNTF, 0.1 ng/ml GDNF, 1 ng/ml BDNF, 1× penicillin-streptomycin) at 37 °C and 5% $CO_2$.

## Lentiviral particle production and motor neuron transduction

LAR shRNA and control viral particles were generated by co-transfecting shRNA, packaging and envelope plasmids into Lenti-X HEK293T cells with Lipofectamine 3000 using manufacturer's instructions. Medium containing lentiviral particles was collected at 48 and 72 h after transfection, concentrated using Lenti-X concentrator and resuspended in Opti-MEM media. Viral particles were stored at −80 °C until use. Neurons were transduced on day in vitro (DIV) 3 by adding viral particles directly to the medium. After 48 h (DIV5), motor neurons were either lysed for western blot analyses or immunostained.

## $H_CT$ preparation and labelling

HA-$H_CT$, VSVG-$H_CT$ and $H_CT$ fused to a cysteine-rich tag were prepared as previously described (Restani et al, 2012). HA-$H_CT$ was used for immunofluorescence, while VSVG-$H_CT$ for immunoprecipitation experiments. $H_CT$ fused to a cysteine-rich tag was labelled with AlexaFluor 555 C2 maleimide or AlexaFluor 647 C2 maleimide following manufacturer's instructions and used in direct immunofluorescence assays.

## Cell-based assays

General immunofluorescence was carried out as described below. Ventral horn cultures at DIV5, after appropriate treatment and incubation, were cooled on ice. Surface-bound probes were removed by washing the cells in mildly acidic buffer (0.2 M acetic acid, 0.5 M NaCl, pH 2.4) for 1 min on ice. After a cold PBS wash, cells were fixed (4% paraformaldehyde, 5% sucrose in PBS), permeabilized and blocked (10% horse serum, 0.5% bovine serum albumin and 0.2% Triton X-100 in PBS) for 10 min at room temperature. Cells were then stained with primary and fluorescently labelled secondary antibodies in blocking buffer for 1 h each at room temperature and mounted. Note that rat anti-PTPRδ and goat anti-ChAT antibodies were incubated overnight at 4 °C. All primary and secondary antibodies were used at 1:500 and 1:1000, respectively, with the exception of the anti-ChAT antibody which was used at 1:50.

For surface expression/localisation analyses, N2a cells were transfected at DIV1. After 24 h, cells were washed with PBS and treated with an anti-HA or anti-nidogen antibody in blocking buffer (without Triton X-100) for 1 h at room temperature. After another wash with PBS, cells were fixed, permeabilized and taken forward for immunostaining using AlexaFluor 488-conjugated anti-rat or anti-rabbit antibodies. Nuclei were stained by supplementing the secondary antibody solution with 0.5 μg/ml 4′,6-diamidino-2-phenylindole (DAPI).

For co-localisation and correlation experiments, motor neurons plated on poly-L-ornithine and iMatrix 511-coated coverslips were incubated with 40 nM $H_C$T-555/$H_C$T-647 and an anti-nidogen-2 antibody for 1 h at 37 °C. After acid washing, fixation and permeabilization, cells were stained with primary antibodies against LAR/PTPRδ and βIII tubulin, which were revealed using appropriate secondary antibodies.

For LAR knockdown, ventral horn cultures were transduced with lentiviral particles on DIV3. Motor neurons transduced with LAR shRNA#2-expressing lentiviruses, were magnetofected with 0.5 μg of pDisplay-HA-eLAR-Myc plasmid on DIV4 to express the HA-eLAR-Myc fusion protein, thus rescuing endogenous LAR depletion. On DIV5, cultures were incubated with 25 nM $H_C$T-647 for 1 h at 37 °C, acid washed and then stained as described above. Anti-βIII tubulin, anti-GFP and anti-HA antibodies were used for immunodetection, which were revealed using fluorescently labelled secondary antibodies.

To check the relevance of TrkB activity on $H_C$T internalisation, motor neuron media was replaced by Neurobasal for 1 h. Motor neuron media was supplemented with 100 nM PF-06273340 and an anti-BDNF blocking antibody (1:50) before being added back to the cells. After 30 min, 25 nM HA-$H_C$T was added, incubated for 1 h at 37 °C, followed by acid washing and immunostaining using anti-HA and βIII tubulin antibodies. Primary antibodies were revealed using AlexaFluor 488-conjugated anti-rat and AlexaFluor 647-conjugated anti-rabbit antibodies, respectively.

For $H_C$T uptake experiments in the presence of LAR/PTPRδ FNIII domains and fragments, recombinant proteins were cloned, expressed in bacteria and purified (Vilstrup et al, 2020). Inclusion of the 8th FNIII domain of LAR or PTPRδ caused an aggregation of the recombinant protein, hence it was omitted. PTPRδ Ig1-3 fragment was expressed and purified from Expi293F cells, using established protocols. Purified proteins were dialyzed (50 mM Tris-Cl pH 7.4, 300 mM NaCl, 5% glycerol) and stored at −20 °C. Motor neuron cultures were pre-incubated with the indicated concentrations of recombinant proteins for 30 min on ice. Cells were then pulsed with 40 nM $H_C$T-555 and a nidogen-2 antibody for 10 min. After replacement of media with fresh motor neuron media, cultures were shifted to 37 °C and allowed to internalise the nidogen-$H_C$T complex for 45 min, followed by acid washing and immunodetection using a mouse anti-βIII tubulin antibody. DyLight 405-conjugated anti-mouse and AlexaFluor 488-conjugated anti-rabbit secondary antibodies were used to reveal the cellular localisation of βIII tubulin and internalised nidogen-2, respectively.

## Image acquisition and analysis

All images were acquired using an inverted Zeiss LSM 780 (with a 40×, 1.3 NA DIC Plan-Apochromat oil-immersion objective) or Zeiss LSM 980 (with a 40×, 1.3 NA DIC Plan-Neofluar oil-immersion objective) confocal microscopes. Co-localisation experiments were imaged on a Zeiss LSM 980 microscope in Airy-scan mode, using a 63×, 1.4 NA DIC Plan-Apochromat oil-immersion objective. All images pertaining to a dataset were acquired using the same microscope settings, processed with Fiji, and, when appropriate, scaled using the same settings. Maximum intensity-projected z-stacks were used for image analysis, which was performed in SynPAnal (Danielson and Lee, 2014). βIII tubulin-positive neurites were manually selected and integrated intensities of internalised $H_C$T and nidogen-2 per unit of axonal length were quantified. In LAR knockdown experiments, GFP was used as a marker of lentiviral transduction, hence neurites positive for both GFP and βIII tubulin were selected. In rescue experiments with recombinant HA-eLAR-Myc, neurites positive for HA, GFP and βIII tubulin were selected for analysis. Co-localisation and correlation analysis were performed in Fiji using in-built Plot Profile and intensity measurement functions, respectively. In correlation analysis, 3–4 neurites of 35–60 μm length were randomly selected in each image in the βIII tubulin channel and corresponding integrated intensities of $H_C$T, LAR/PTPRδ and nidogen-2 were calculated.

## Cell lysis and immunoprecipitation

Relevant plasmids were transfected in N2a cells after 24 h of plating using Lipofectamine 3000 and manufacturer's protocols. After 6 h, culture media was replaced with differentiation media (DMEM, 1% FBS, 1% GlutaMAX, 10 μM retinoic acid). After 48–72 h, cells were treated with 80 nM VSVG-$H_C$T for 10 min at 37 °C, and then lysed in immunoprecipitation buffer (20 mM Tris-Cl pH 8.0, 137 mM NaCl, 10% glycerol, 0.5% NP-40, protease and phosphatase inhibitors) for 20 min on ice. Cell debris was removed by centrifuging the lysates at 21,000×g for 15 min at 4 °C. Two types of beads were used for immunoprecipitation experiments: (i) protein A or G sepharose beads non-covalently bound to appropriate antibodies for 2 h at 4 °C (Fig. 1; Appendix Fig. S1); or, (ii) magnetic beads covalently conjugated to an anti-HA antibody (Figs. 4 and 7). Lysates were incubated with beads for 2 h at 4 °C under gentle agitation, washed 3–5 times in immunoprecipitation buffer and then treated with 4× Laemmli buffer at 95 °C for 4 min. Samples were analysed by western blotting. Relevant

controls were used for each experiment, as described in the figure legends. 5% lysate was loaded as input.

Motor neuron lysates were prepared in RIPA buffer (50 mM Tris-Cl pH 7.5, 150 mM NaCl, 1% NP-40, 0.5% sodium deoxycholate, 0.1% SDS, 1 mM EDTA, 1 mM EGTA, protease and phosphatase inhibitors) as described above and analysed using western blotting. Total protein was probed using Coomassie R-250 staining on membranes. All blots were imaged on the ChemiDoc™ Touch Imaging System (Bio-Rad Laboratories); data analysis was performed using the ImageLab software (Bio-Rad Laboratories).

## Enzyme-linked immunosorbent assays

1 mg/ml anti-nidogen-2 capture antibody, diluted in PBS, was applied to wells, gently agitated for 5 min at room temperature, and incubated overnight at 4 °C. Wells were blocked using 5% bovine serum albumin (BSA) in PBS containing 0.05% Tween-20 (PBST) for 1.5 h at room temperature. 0.5 picomoles of recombinant nidogen-2 was mixed with varying concentrations of the relevant LAR/PTPRδ domains/fragments (50 nM–70 μM) in binding buffer (20 mM Tris-Cl pH 8.0, 137 mM NaCl, 10% glycerol, 0.5% NP-40) and shaken for 2 h at 4 °C. The mixture was applied to the wells and allowed to bind for 1.5 h at room temperature. After multiple washes using PBST, the bound complex was detected using an anti-FLAG or anti-His primary antibody (1:1000) and an HRP-conjugated anti-mouse secondary antibody (1:5000). All antibodies were diluted in PBST containing 1% BSA and incubated for 1 h at room temperature. 3,3',5,5'-tetramethylbenzidine was used as a substrate and the reaction was stopped using 2 M $H_2SO_4$. Complex formation was assessed by measuring the absorbance at 450 nm in a FLUOstar Omega microplate reader (BMG Labtech). Reactions devoid of any LAR FNIII domain served as background. All readings were normalised to absorbance values obtained at the 5 μM concentration.

## Footprint and gait analysis

Age- and body weight-matched C57Bl6/SJ6 female mice were randomly assigned to experimental groups and then injected intramuscularly with a sub-lethal dose of TeNT (0.5 ng/kg) alone, or TeNT mixed with LAR and/or PTPRδ fragments. Fragments were injected at the following concentrations: (i) LAR FNIII1-4-FLAG at 56 μM, (ii) LAR FNIII5-7-FLAG at 68 μM, (iii) PTPRδ Ig1-3-His at 20 μM, and (iv) PTPRδ FNIII5-7-FLAG at 100 μM. Protein concentrations in each solution were made up to 10 mg/ml using BSA, so as to rule out spurious results due to differences in protein load. Solutions were injected through the skin and into the right gastrocnemius muscle of isoflurane-anaesthetised mice using pulled, glass micropipettes (Drummond Scientific, 5-000-1001-X10), following a published protocol (Sleigh et al, 2020). Mice were monitored for a period of 96 h unless they reached the humane endpoint first (appearance of moderate symptoms: hunched back and paralysis of rear limbs, or disappearance of the righting reflex for 30 s). Footprint assays were performed as previously described (Moritz et al, 2019). Briefly, all mice were trained daily for a week prior to the start of the experiment. At the relevant time post-injection, the hind paws were coloured red, while the ipsilateral (right) front paw was painted black. All colours were non-toxic and water-based. Each mouse was then placed at the start of a narrow tunnel and allowed to walk on paper towards a goal chamber. After retrieval of the mouse, it was returned to its home cage and the testing area was wiped down with ethanol between tests. To quantify the extent of paralysis, two distances were measured: (i) distance between the ipsilateral front and hind paw prints and, (ii) distance between the footprints of the injected and un-injected hindlimbs. Mice with severe plantar flexion were unable to use the injected hindlimb while walking, resulting in the absence of a footprint. When the footprint of the injected limb was absent in more than half the corresponding ipsilateral front pawprints or the contralateral hind pawprints, a value of ∞ (infinite) or 0 was assigned to the coordination distance and step width, respectively. Data were analysed using Fiji. All researchers involved in the experiment were blinded until the end of data analysis.

## Statistical analyses

Data were tested for normality using the Kolmogorov–Smirnov test, while equal variance between groups was assumed. Normally-distributed data were statistically analysed using unpaired $t$-test or one-way analysis of variance (ANOVA) followed by Dunnett's multiple comparisons test. Time course in vivo experiments were analysed using two-way ANOVA, followed by Dunnett's multiple comparisons test. Non-normally distributed data were analysed using Mann–Whitney test or Kruskal–Wallis test followed by Dunn's post-hoc test. Means ± standard error of the mean were plotted. All tests were two-sided and an α-level of $P < 0.05$ was used to determine significance. $P$ values obtained from $t$-test and Mann–Whitney test are indicated in the figures. $P$ values obtained from ANOVA are indicated in the figure legends, while those obtained from post-hoc tests are indicated in the figures. GraphPad Prism 9 software (version 9.5.0) was used for statistical analyses and figure production.

## Ethics statement

All experiments were conducted under the guidelines of the UCL Queen Square Institute of Neurology Genetic Manipulation and Ethics Committees and in accordance with the European Community Council Directive of 24 November 1986 (86/609/EEC). Animal experiments were performed under license from the UK Home Office in accordance with the Animals (Scientific Procedures) Act, 1986 and were approved by the UCL Queen Square Institute of Neurology Ethical Review Committee.

# Data availability

The datasets produced in this study are available in the following databases: Imaging datasets: BIA 10.6019/S-BSST1406 (https://www.ebi.ac.uk/biostudies/studies/S-BSST1406). Western blots: BIA 10.6019/S-BSST1406 (https://www.ebi.ac.uk/biostudies/studies/S-BSST1406). Movies: BIA 10.6019/S-BSST1406 (https://www.ebi.ac.uk/biostudies/studies/S-BSST1406). ELISA: BIA 10.6019/S-BSST1406 (https://www.ebi.ac.uk/biostudies/studies/S-BSST1406).

The source data of this paper are collected in the following database record: biostudies:S-SCDT-10_1038-S44318-024-00164-8.

## Peer review information

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

## Acknowledgements

We thank Prof. Carlo Sala (University of Milan, Italy) for LAR, PTPRδ and PTPRσ plasmids, Dr. Darius Vasco Köster (Warwick University, UK) and Prof. Ornella Rossetto (University of Padova, Italy) for scientific input and critical reading of the manuscript, and Simon L. Duerr (École Polytechnique Fédérale de Lausanne, Switzerland) for generously sharing high-quality science illustrations *via* https://bioicons.com/, which were used for rendering schematics in this manuscript. We thank the personnel of the Denny Brown Laboratories (Queen Square Institute of Neurology, University College London, UK) for maintenance of mouse colonies. The synopsis figure for this paper was made using BioRender. This project was funded by a Human Frontier Science Program Long-Term Fellowship LT000220/2017-L (SS), UKRI Medical Research Council awards MR/S006990/1 and MR/Y010949/1 (JNS), UCL Therapeutic Acceleration Support scheme supported by funding from UKRI MRC IAA 2021 UCL MR/X502984/1 (JNS), UCL Neurogenetic Therapies Programme funded by The Sigrid Rausing Trust (JNS, GS), Wellcome Trust Awards 107116/Z/15/Z and 223022/Z/21/Z (GS), UK Dementia Research Institute Foundation Award UKDRI-1005 (GS), and an Alzheimer's Society PhD Studentship Grant 520 (CP, GS).

## Author contributions

**Sunaina Surana**: Conceptualization; Resources; Data curation; Formal analysis; Supervision; Funding acquisition; Validation; Investigation; Visualization; Methodology; Writing—original draft; Project administration; Writing—review and editing. **David Villarroel-Campos**: Investigation; Writing—review and editing. **Elena R Rhymes**: Investigation; Writing—review and editing. **Maria Kalyukina**: Investigation. **Chiara Panzi**: Funding acquisition; Investigation. **Sergey S Novoselov**: Validation. **Federico Fabris**: Investigation. **Sandy Richter**: Resources. **Marco Pirazzini**: Investigation; Writing—review and editing. **Giuseppe Zanotti**: Investigation; Writing—review and editing. **James N Sleigh**: Funding acquisition; Investigation; Writing—review and editing. **Giampietro Schiavo**: Conceptualization; Supervision; Funding acquisition; Project administration; Writing—review and editing.

Source data underlying figure panels in this paper may have individual authorship assigned. Where available, figure panel/source data authorship is listed in the following database record: biostudies:S-SCDT-10_1038-S44318-024-00164-8.

## Disclosure and competing interests statement

The authors declare no competing interests.

# Expanded View Figures

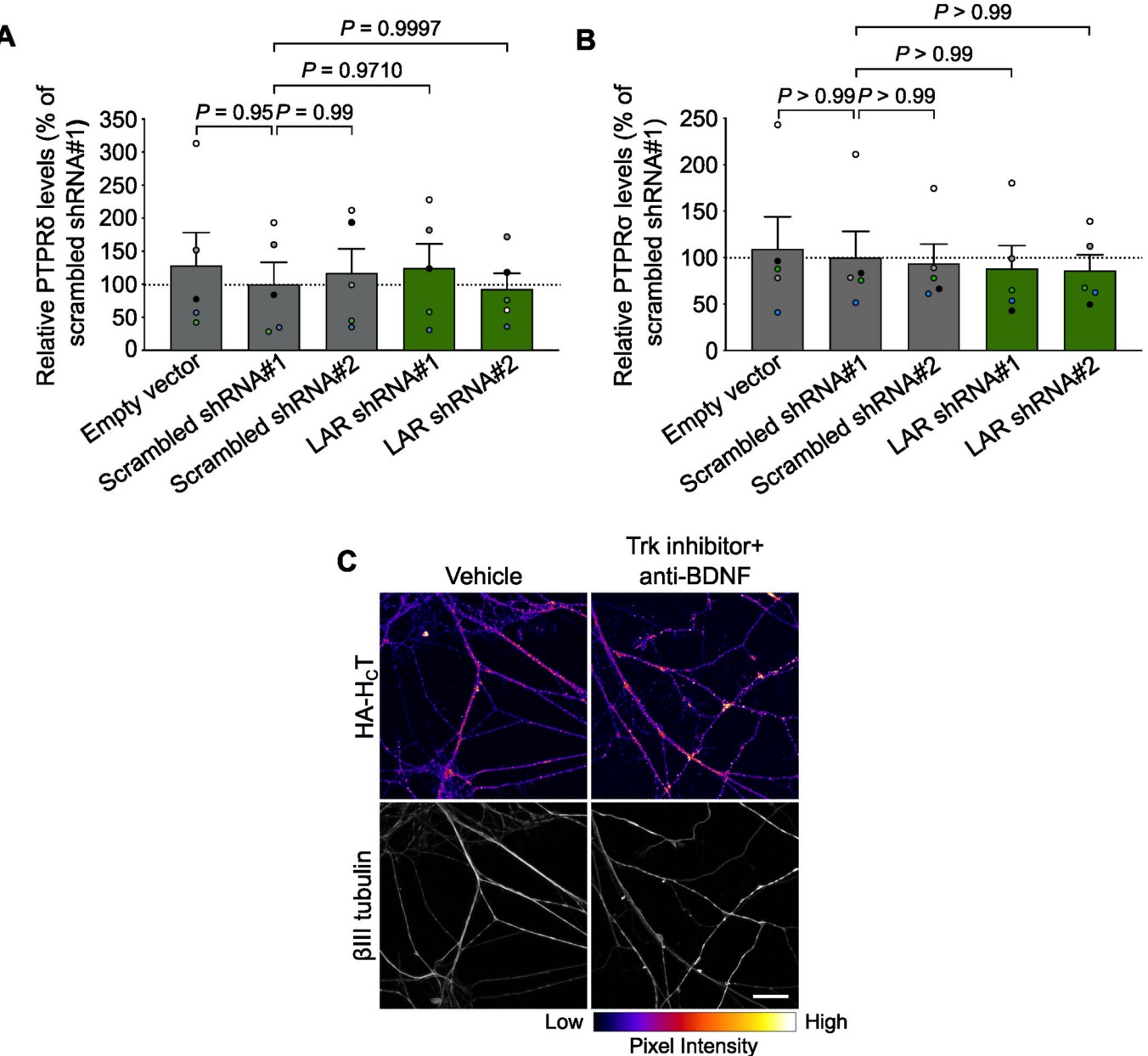

**Figure EV1. Lentivirus-mediated LAR knockdown has no effect on expression levels of PTPRδ and PTPRσ.**

(A) PTPRδ quantification in lysates of ventral horn cultures transduced with lentiviruses encoding shRNAs against mouse LAR. Results were tested for statistical significance using one-way ANOVA ($P = 0.9480$), followed by Dunnett's multiple comparisons test ($n = 5$ independent experiments; error bars indicate s.e.m.). (B) PTPRσ quantification in lysates of ventral horn cultures transduced with lentiviruses encoding shRNAs against mouse LAR. Results were tested for statistical significance using Kruskal–Wallis test ($P = 0.9665$), followed by Dunn's *post-hoc* test ($n = 5$ independent experiments; error bars indicate s.e.m.). Data are presented as a percentage of the total levels of PTPRδ (A) and PTPRσ (B) in neurons treated with scrambled shRNA#1. (C) Representative confocal images of endocytosed HA-H$_C$T in motor neurons treated with the pan-Trk inhibitor PF-06273340 and an anti-BDNF antibody. Images in the top panel have been colour mapped based on their intensities. Vehicle refers to DMSO-treated cultures. Scale bar: 20 μm.

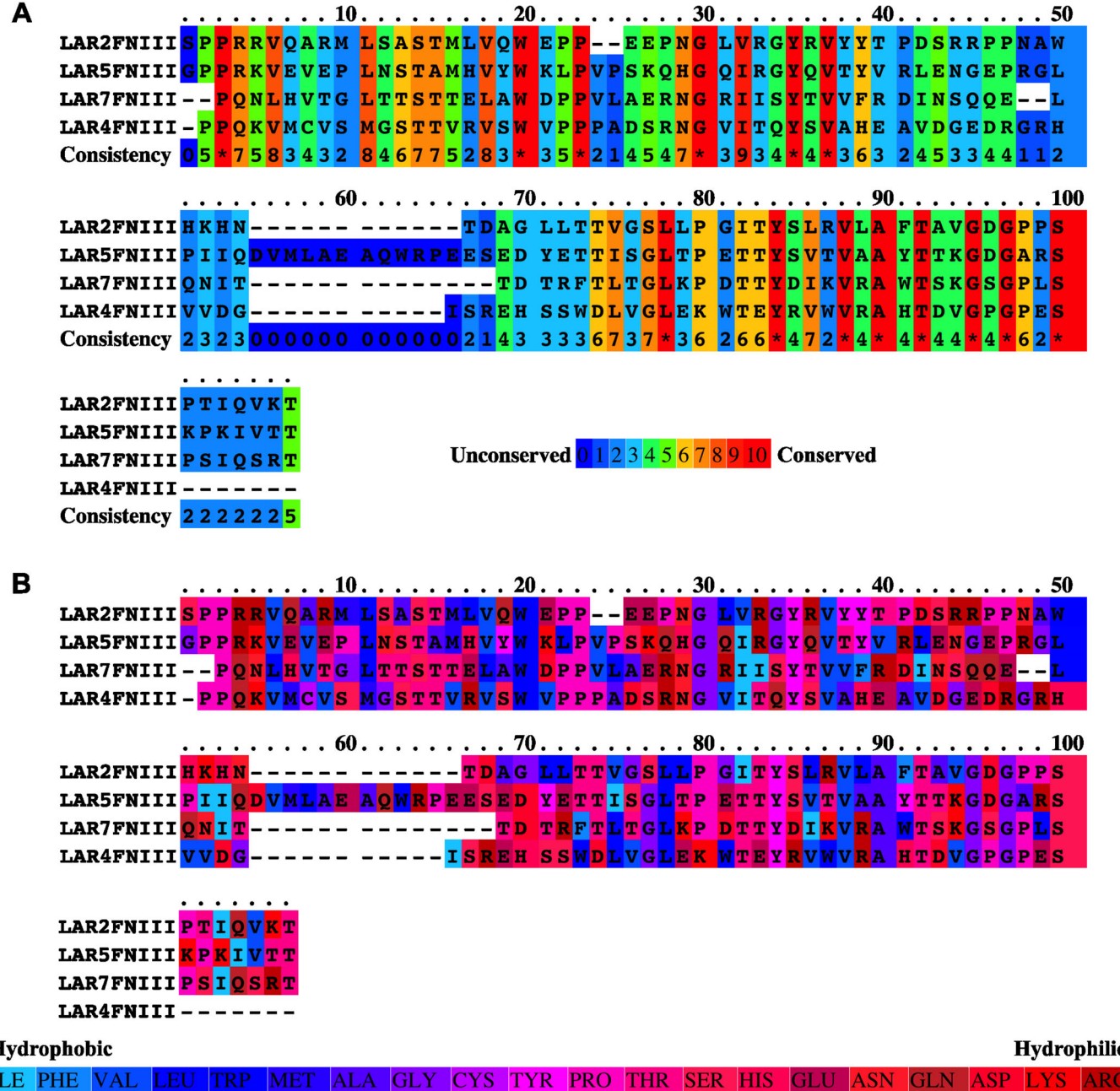

**Figure EV2.  Sequence alignment of the human LAR FNIII2, FNIII4, FNIII5 and FNIII7 domains using PRALINE.**

(A) Sequence alignment of the human LAR FNIII2, FNIII4, FNIII5 and FNIII7 domains. Scores range from 0 for the least conserved alignment position, up to 10 for the most conserved position. (B) Conservation of hydrophobicity/hydrophilicity in the nidogen-binding domains of LAR. Colour assignments from hydrophobic to hydrophilic are shown below the alignment.

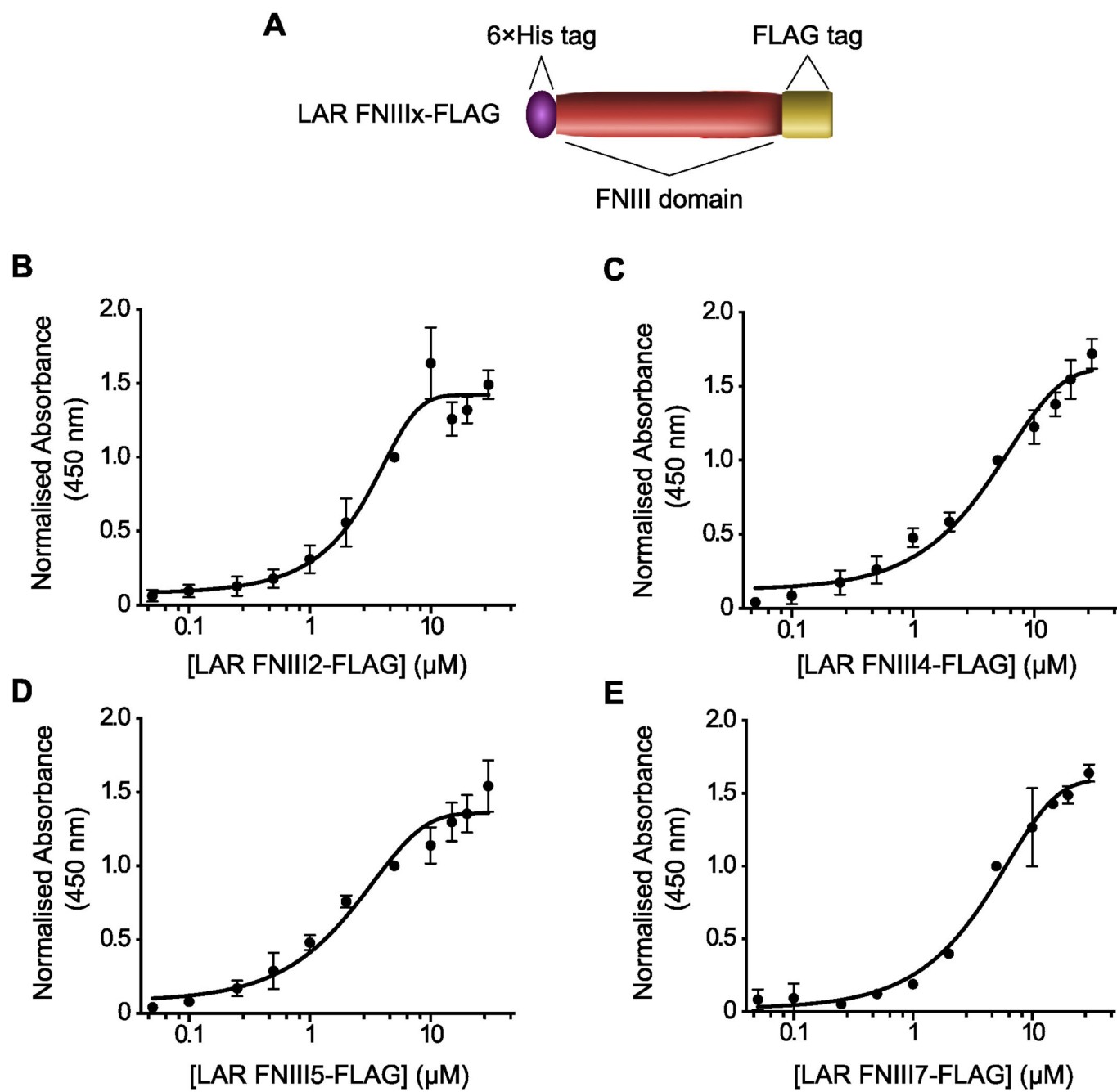

**Figure EV3. Dose-dependence of the interaction between nidogen-2 and soluble LAR FNIII2, FNIII4, FNIII5 and FNIII7 domains.**

(A) Schematic of LAR FNIII fusion proteins used for bacterial expression and purification. Each nidogen-binding FNIII domain was tagged with a 6×His tag at the N-terminus and a FLAG tag at the C-terminus. (B–E) Plots showing in vitro binding between purified nidogen-2 and bacterially expressed LAR FNIII2-FLAG (B), LAR FNIII4-FLAG (C), LAR FNIII5-FLAG (D), and LAR FNIII7-FLAG (E). Serial dilutions of each purified LAR FNIII domain (50 nM–30 μM) were added to a fixed amount of immobilised nidogen-2 (0.5 picomoles), followed by addition of an anti-FLAG antibody to reveal complex formation using ELISA. All datapoints were normalised to the absorbance obtained using 5 μM of LAR FNIII domain ($n = 3$ independent experiments; error bars indicate s.e.m.).

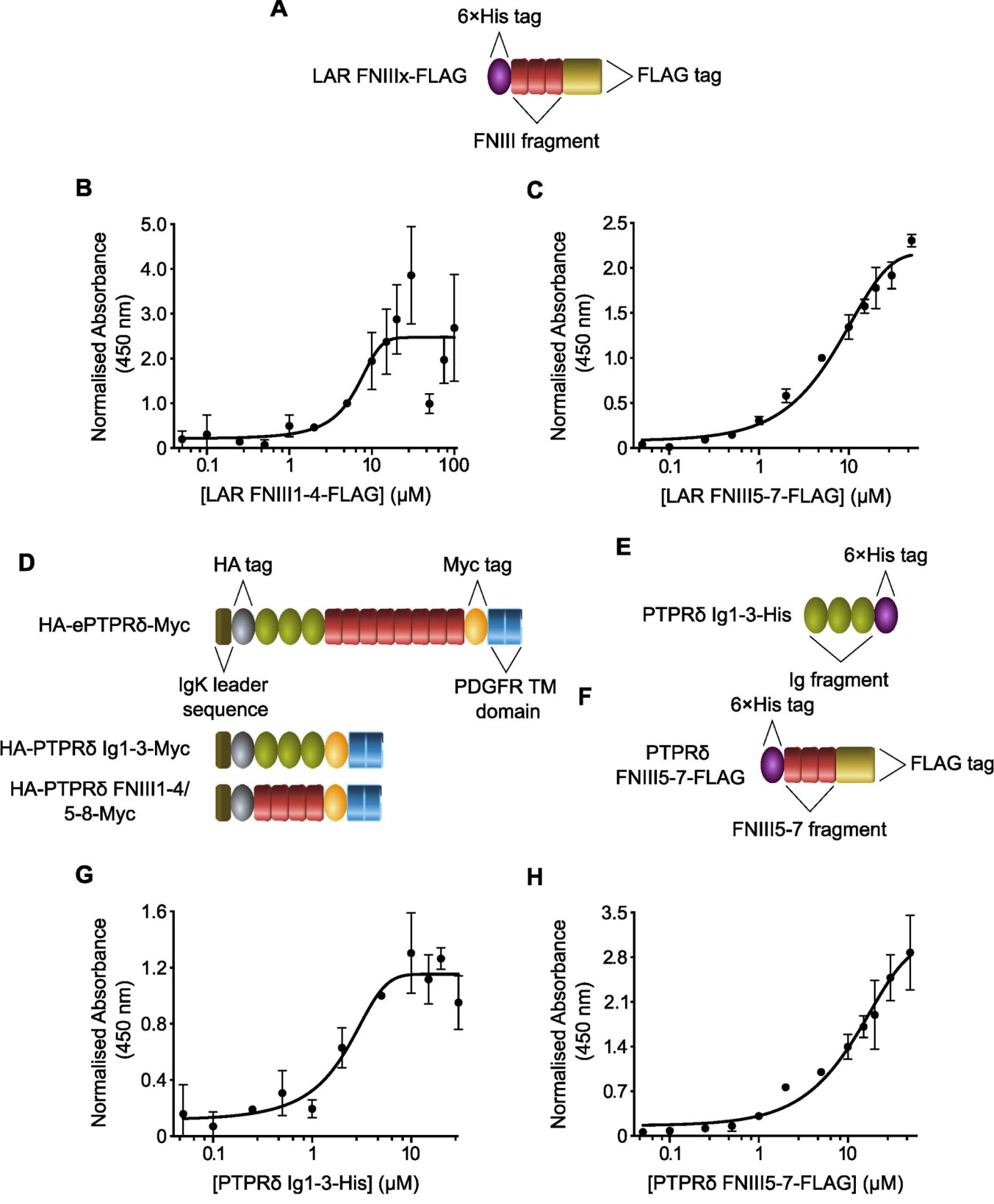

◄ **Figure EV4. Nidogen-2 and LAR/PTPRδ fragments exhibit dose-dependent interactions in vitro.**

(A) Schematic of recombinant LAR FNIII fragments used for bacterial expression and purification. LAR FNIII1-4 and FNIII5-7 fragments were tagged with a 6×His tag at the N-terminus and a FLAG tag at the C-terminus. (B, C) Plots showing in vitro binding between purified nidogen-2 and bacterially expressed LAR FNIII1-4-FLAG (B) and LAR FNIII5-7-FLAG (C). Serial dilutions of each purified LAR fragment were added to a fixed amount of immobilised nidogen-2 (0.5 picomoles), followed by addition of an anti-FLAG antibody to reveal complex formation using ELISA. All datapoints were normalised to the absorbance obtained using 5 µM of LAR FNIII fragment ($n = 3$ independent experiments; error bars indicate s.e.m.). (D) Schematic of PTPRδ fragments used to identify the interacting domains between PTPRδ and nidogens. Truncated proteins were fused to the murine Igκ-chain leader sequence and an HA tag at the N-terminus; the C-terminus was fused to the PDGFR transmembrane domain and a Myc tag. (E, F) Schematic of recombinant PTPRδ fragments used for protein expression and purification. PTPRδ Ig1-3 was tagged with a 6×His tag at the C-terminus (E), while the FNIII5-7 fragment was tagged with a 6×His tag at the N-terminus and a FLAG tag at the C-terminus (F). (G, H) Plots showing in vitro binding between purified nidogen-2 and recombinant PTPRδ Ig1-3-His (G) and PTPRδ FNIII5-7-FLAG (H). Serial dilutions of each PTPRδ fragment were added to a fixed amount of immobilised nidogen-2 (0.5 picomoles), followed by addition of an anti-His or anti-FLAG antibody, respectively, to reveal complex formation using ELISA. All datapoints were normalised to the absorbance obtained using 5 µM of PTPRδ FNIII fragment ($n = 3$ independent experiments; error bars indicate s.e.m.).

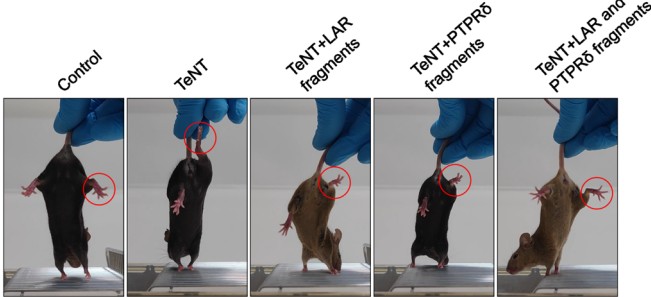

**Figure EV5.  Soluble LAR and PTPRδ fragments rescue posture defects in TeNT-injected mice.**

Representative still images showing the outcomes of a tail suspension assay in mice injected with sub-lethal doses of TeNT alone, or TeNT pre-mixed with LAR and/or PTPRδ fragments. Mice were injected in the gastrocnemius muscle of the right hindlimb (red circles); non-injected mice were used as a negative control.

