## [Peer Review File · The EMBO Journal]

The tyrosine phosphatases LAR and PTPR δ act as receptors of the nidogen-tetanus toxin complex

Sunaina Surana, David Villarroel-Campos, Elena Rhymes, Maria Kalyukina, Chiara Panzi, Sergey Novoselov, Federico Fabris, Sandy Richter, Marco Pirrazini, Giuseppe Zanotti, James Sleight, and Giampietro Schiavo

Corresponding author(s): Giampietro Schiavo (giampietro.schiavo@ucl.ac.uk) , Sunaina Surana (s.surana@ucl.ac.uk)

Review Timeline:

Submission Date:	3rd Feb 23
Editorial Decision:	23rd Mar 23
Appeal Received:	19th Apr 24
Editorial Decision:	28th May 24
Revision Received:	14th Jun 24
Accepted:	19th Jun 24

Editors: Karin Dumstrei and Ioannis Papaioannou

Transaction Report:

Dear Gipi,

Thank you for sending me your point-by-point response to the referees' comments. I have now had a chance to take a look at the response and discussed it with my colleagues.

I appreciate that you can address some of the raised issues, but I am also afraid that the analysis doesn't go far enough to consider publication here.

I completely see the issue regarding adding in vivo data, but we would need some insight along those lines to consider publication here.

I am very sorry that I can't be more positive on this occasion.

Yours sincerely,

Karin

Karin Dumstrei, PhD
Senior Editor
The EMBO Journal

Referee #1:

-In this article, the authors built upon their previous seminal finding that tetanus neurotoxin exploits nidogen-1 and -2 for its binding to motor neurons, subsequent internalization, and axonal retrograde transport (<https://doi.org/10.1126/science.1258138>). Here they identified the tyrosine phosphatase LAR as the nidogen-TeNT surface receptor. The question tackled by the authors is of extremely high relevance in neuroscience, and cell biology and for the clinics.

-The experiments shown here were well carried out and controlled. However, they were all conducted in cultured neurons. In their previous study on nidogen, the authors were able to obtain important evidence in vivo in the animal. As pointed out by the authors, Thy-1 was previously proposed as a TeNT receptor and this claim was contradicted by experiments in the Thy-1 KO mouse. For the same reason, it appears quite necessary to have a minimal set of data in an animal model in the case of LAR. Given the current state of data shown here, this goal appears achievable.

-Minor comments: the authors cite page 7 unpublished results of PLA: these should be shown.

Referee #2:

This study reports the role of the LAR receptor of protein tyrosine phosphatases in the regulation of the entry of the TeNT-nidogen complex into motor nerve terminals. In support of this, the authors show LAR binds to TeNT through specific FNIII domains. Knockdown of LAR inhibits the entry of TeNT-nidogen into motor neurons. In addition, recombinant FNIII domains inhibit TeNT-nidogen entry into motor neurons. These effects are independent of the TrkB modulatory effect of LAR.

This study is carefully designed and performed. The results convincingly suggest LAR as a novel mediator of the entry of TeNT-nidogen entry into motor neurons. Given that LAR family presynaptic adhesion molecules play critical roles in synapse development and function, this study adds one more layer to the known functions of LAR family proteins, suggesting they have both physiological and toxin-related roles.

1. In Figure 1, the authors test LAR, PTPRdelta, and PTPRsigma for TeNT-nidogen binding and find that LAR and PTPRdelta bind TeNT-nidogen but PTPRsigma does not, which is surprising considering the high similarity in the domain organization and FNIII sequences across LAR family proteins. I wonder what could be the reasons. Are the FNIII domains of LAR and PTPRdelta more similar to each other than the FNIII domains in PTPRsigma in terms of aa sequence, key residue, or domain structure?

2. The authors used individual FNIII domains or their mixtures to inhibit the motor neuron entry of TeNT-nidogen. Although the modular nature of FNIII domains is appreciated, it is unclear why the author used a mixture of FNIII domains rather than naturally occurring FNIII domains such as FN 1-4 or FN5-8, which may be more competitive in inhibiting the TeNT-nidogen entry. In addition, Figure 7 should also contain data from the use of a negative control (FNIII5 at an equimolar concentration).

3. I wonder if the phosphatase domains of the LAR family protein affect the interaction of LAR with TeNT-nidogen. Does the phosphatase activity have anything to do with the LAR-dependent regulation of TrkB activity?
4. It is unclear which members of the LAR family proteins are expressed in motor neurons. Approaches such as in situ hybridization may help clarify this issue.
5. Presynaptic LAR family proteins are known to interact with several postsynaptic adhesion molecules through various domains. I wonder whether there is any possibility of interference between the known transsynaptic interactions and the FNIII-TeNT/nidogen interaction.
6. The LAR panel in Figure 3A should be replaced with a better one.

** As a service to authors, EMBO Press provides authors with the possibility to transfer a manuscript that one journal cannot offer to publish to another EMBO publication or the open access journal Life Science Alliance launched in partnership between EMBO Press, Rockefeller University Press and Cold Spring Harbor Laboratory Press. The full manuscript and if applicable, reviewers' reports, are automatically sent to the receiving journal to allow for fast handling and a prompt decision on your manuscript. For more details of this service, and to transfer your manuscript please click on Link Not Available. **

Response to Reviewers' Comments

We thank the Reviewers for their appreciative comments and the editor for inviting us to respond to their suggestions aiming at improving our manuscript. All comments are addressed in the point-by-point response below, with our responses in blue. Changes to the main text are in green.

Referee #1

1. In this article, the authors built upon their previous seminal finding that tetanus neurotoxin exploits nidogen-1 and -2 for its binding to motor neurons, subsequent internalization, and axonal retrograde transport (<https://doi.org/10.1126/science.1258138>). Here they identified the tyrosine phosphatase LAR as the nidogen-TeNT surface receptor. The question tackled by the authors is of extremely high relevance in neuroscience, and cell biology and for the clinics.

We thank the Reviewer for their positive comments as well as for their assessment that our study has broad implications for molecular neuroscience, cell biology and clinical toxicology.

2. The experiments shown here were well carried out and controlled. However, they were all conducted in cultured neurons. In their previous study on nidogen, the authors were able to obtain important evidence *in vivo* in the animal. As pointed out by the authors, Thy-1 was previously proposed as a TeNT receptor and this claim was contradicted by experiments in the Thy-1 KO mouse. For the same reason, it appears quite necessary to have a minimal set of data in an animal model in the case of LAR. Given the current state of data shown here, this goal appears achievable.

We thank the reviewer for raising this point. We agree that *in vivo* validation is a key step in understanding the role of potential membrane receptors in TeNT intoxication. However, this goal is complicated by the functional redundancy between LAR and other RPTPs. Our experiments have consistently shown that knockdown of endogenous LAR or disruption of the LAR-nidogen interaction leads to a ~40-45% reduction in H_cT uptake. The incomplete block of H_cT internalisation under these conditions indicated that the nidogen-TeNT complex relies on additional membrane proteins for its endocytosis. Based on these considerations, we expected that any *in vivo* intervention, including knocking out LAR or inhibiting the LAR-nidogen interaction, would only result in a ~50% reduction in TeNT internalisation. Because TeNT is highly toxic even at very low concentrations (Surana *et al*, 2018), a ~50% reduction in its uptake is still expected to induce spastic paralysis, and possibly death, in injected mice.

To circumvent these issues, we decided to further explore the role of PTPR δ in the internalisation of the nidogen-H_cT complex in motor neurons. Our experiments indeed showed that PTPR δ is a binding partner of nidogens (Fig 1F-H of the revised manuscript). Using a truncation and co-immunoprecipitation approach, we found that nidogens bind to the Ig1-3 and FNIII5-8 domains of PTPR δ . These fragments were cloned, purified, and the resulting recombinant proteins used in competitive inhibition assays similar to those performed with the LAR FNIII domains in the submitted manuscript. We found that when neuronal cultures were treated with H_cT-555 and an anti-nidogen-2 antibody in the presence of an excess of PTPR δ Ig1-3-His and FNIII5-7-FLAG, there was a ~30-40% reduction in the internalisation of the nidogen-H_cT complex (Fig R1A, B). These findings suggested that PTPR δ is a component of the H_cT receptor complex.

In parallel, we combined the nidogen-binding FNIII domains of LAR into two recombinant fragments, namely the FNIII1-4-FLAG and FNIII5-7-FLAG fragments. When motor neurons were treated with an excess of these fragments, they were able to collectively inhibit nidogen-H_cT uptake by ~35% (Fig R1C, D). Remarkably, inclusion of the 8th FNIII domain of LAR or PTPR δ caused an aggregation of the recombinant protein, hence it was omitted from the fragments.

Figure R1. Nidogen-binding fragments derived from LAR and PTPR δ inhibit uptake of the nidogen-HcT complex. (A, B) Plots showing internalised nidogen-2 (A) and HcT-555 (B) in motor neurons treated with 0.2 μ M of PTPR δ Ig1-3-His and 1 μ M of PTPR δ FNIII5-7-FLAG mixed together. **(C, D)** Quantification of endocytosed nidogen-2 (C) and HcT-555 (D) in the presence of 0.56 μ M of LAR FNIII1-4-FLAG and 0.68 μ M of LAR FNIII5-7-FLAG mixed together. Results were analysed for statistical significance using an unpaired *t*-test in (A-C) and Mann-Whitney test in (D). Data are presented as a percentage of internalised nidogen-2 or HcT in buffer-treated controls ($n = 3$ independent experiments; error bars indicate s.e.m.). These data are now included in Figs 6 and 7 of the revised manuscript.

We then tested the ability of these fragments to inhibit binding and uptake of the nidogen-TeNT complex *in vivo*. Briefly, mice were injected with sub-lethal doses of full-length TeNT either alone or in combination with LAR and PTPR δ fragments. Injections were carried out in the gastrocnemius muscle and the resulting local spastic paralysis was assessed. Gait coordination was monitored by measuring the distance of the injected hind paw from the footprint of the ipsilateral fore paw (**Fig R2A**; black bracket in control panel) (Bercsenyi *et al*, 2014), whereas step width was quantified by measuring the distance between the injected and non-injected hind paws (**Fig R2A**; red bracket in the control panel) (Moritz *et al*, 2019).

Mice injected with TeNT alone developed severe gait abnormalities at 96 h post-injection, with permanent plantar flexion of the affected hind paw. This was accompanied by hyperextension and flexion of the injected leg during a tail suspension assay, leading to a significant decrease in gait coordination as well as step width. Co-administration of TeNT with the LAR fragments FNIII1-4-FLAG and FNIII5-7-FLAG produced a marked improvement of gait defects and posture during tail suspension, resulting in the restoration of step width and coordination. Similar observations were made when TeNT was co-injected with an excess of the PTPR δ fragments Ig1-3-His and FNIII5-7-FLAG. To test the possibility of an additive effect of LAR and PTPR δ , mice were administered with TeNT and all four nidogen-binding fragments of these tyrosine phosphatases. This group of mice displayed minimal gait abnormalities, with the injected hindlimb extended away from the midline and splayed normally during tail suspension. The coordination distance and step width of these mice were indistinguishable from control mice, indicating a near-complete prevention of TeNT-induced spastic paralysis (**Fig R2**).

Taken together, our data show that interfering with the endogenous LAR-nidogen and PTPR δ -nidogen interactions block spastic paralysis in mice, confirming that LAR and PTPR δ are indeed the surface receptors for the nidogen-TeNT complex.

Figure R2. Recombinant LAR and PTPR δ fragments inhibit spastic paralysis in mice. (A) Footprints of mice 96 h after injection of TeNT with nidogen-binding LAR and/or PTPR δ fragments. The paw prints of the injected right hindlimb are marked with red circles, while the paw prints of the non-injected hindlimb are marked with dotted red circles. Black circles mark the ipsilateral front paw. The black bracket in the control panel shows coordination length, whereas the red bracket shows the step width between the hind paws. Scale bar: 1 cm. (B, C) Coordination (B) and step width (C) 96 h after injection ($n = 3$ independent experiments; error bars indicate s.e.m.). Results were analysed for statistical significance using one-way ANOVA ($P = 0.0053$ and 0.0507 for (B) and (C), respectively), followed by Dunnett's *post-hoc* test. These data are now included in Fig 8 of the revised manuscript.

3. Minor comments: the authors cite page 7 unpublished results of PLA: these should be shown.

The results of our proximity biotinylation screen, which relied on fusing a mutant biotin ligase BirA moiety to H_CT, are presented in **Fig R3**. In this approach, BirA-H_CT was incubated with embryonic stem cell-derived motor neurons and then allowed to be internalised in signalling endosomes. This led to BirA-dependent biotinylation, and subsequent identification by mass spectrometry, of proteins present within a 10-15 nm radius of the H_CT fusion protein in neuronal endosomes. As shown in **Fig R3**, LAR and nidogen-1 were detected by mass spectrometry upon streptavidin purification of the cell lysates. We are currently in the process of further refining this approach and analysing the data, which will be fully described in a future manuscript.

Given the evolving nature of this set of experiments and our plans to report additional interactors in a future manuscript, we kindly request to include this figure and data only in this response to Reviewers letter, which will be linked to the manuscript, if published.

Referee #2

This study reports the role of the LAR receptor of protein tyrosine phosphatases in the regulation of the entry of the TeNT-nidogen complex into motor nerve terminals. In support of this, the authors show LAR binds to TeNT through specific FNIII domains. Knockdown of LAR inhibits the entry of TeNT-nidogen into motor neurons. In addition, recombinant FNIII domains inhibit TeNT-nidogen entry into motor neurons. These effects are independent of the TrkB modulatory effect of LAR.

This study is carefully designed and performed. The results convincingly suggest LAR as a novel mediator of the entry of TeNT-nidogen entry into motor neurons. Given that LAR family presynaptic adhesion molecules play critical roles in synapse development and function, this study adds one more layer to the known functions of LAR family proteins, suggesting they have both physiological and toxin-related roles.

1. In Figure 1, the authors test LAR, PTPRdelta, and PTPRsigma for TeNT-nidogen binding and find that LAR and PTPRdelta bind TeNT-nidogen but PTPRsigma does not, which is surprising considering the high similarity in the domain organization and FNIII sequences across LAR family proteins. I wonder what could be the reasons. Are the FNIII domains of LAR and PTPRdelta more similar to each other than the FNIII domains in PTPRsigma in terms of aa sequence, key residue, or domain structure?

We are thankful to the Reviewer for this comment. We have now compared the amino acid sequences of the FNIII domains of (i) PTPR δ vs. LAR, and (ii) PTPR σ vs. LAR. Sequence alignments, which are included in the Appendix, indicate that the FNIII domains of these three proteins display similar levels of homology: whereas the FNIII domains of PTPR δ and LAR are 77% similar, PTPR σ and LAR share 69% homology. Their domain structure and organisation are also similar, as predicted by AlphaFold.

Despite their overall similarity, LAR, PTPR δ and PTPR σ have been shown to display distinct interaction profiles (Takahashi & Craig, 2013; Um & Ko, 2013), which include unique and shared ligands across PTPRs (**Table RT1**).

Table RT1. Synaptic binding partners of LAR-RPTPs.

Ligand	Binding LAR-RPTP
NGL-3	LAR/PTPR δ /PTPR σ
TrkB	LAR/PTPR δ
TrkC	PTPR σ
IL1RAPL1	PTPR δ
IL1RAcP	LAR/PTPR δ /PTPR σ
Slitrk1,2,4-6	PTPR σ
Slitrk3	PTPR δ
HSPGs	LAR
CSPGs	LAR/PTPR σ

Accordingly, we observed that whilst nidogens bind exclusively to the FNIII domains of LAR (**Fig 4** of the revised manuscript), the interaction with PTPR δ involves the Ig1-3 and FNIII5-8 domains (**Fig R4**). These results indicate that nidogen-RPTP binding cannot be explained by sequence similarities alone. Furthermore, the extracellular domains of LAR-RPTPs undergo alternative splicing, which controls their ligand-binding specificity. In this study, we have used the full-length extracellular domain of LAR-RPTPs to probe their binding to nidogens,

therefore we are unable to rule out that a splice variant of PTPR σ might exhibit binding to nidogens.

Figure R4. Specificity of the interaction between PTPR δ and nidogen-2. Co-immunoprecipitation and western blot analysis of the binding between HA-PTPR δ Ig1-3-Myc, HA-PTPR δ FNIII1-4-Myc and HA-PTPR δ FNIII5-8-Myc and nidogen-2, in the presence of VSVG-H_cT. Immunoprecipitation was performed using an anti-HA antibody, and samples were probed using an anti-nidogen-2 antibody. The HA-ePTPR δ -Myc fusion protein was used as a positive control and 5% input was loaded. We have now included this panel in Fig 7 of the revised manuscript.

2. The authors used individual FNIII domains or their mixtures to inhibit the motor neuron entry of TeNT-nidogen. Although the modular nature of FNIII domains is appreciated, it is unclear why the author used a mixture of FNIII domains rather than naturally occurring FNIII domains such as FN 1-4 or FN5-8, which may be more competitive in inhibiting the TeNT-nidogen entry. In addition, Figure 7 should also contain data from the use of a negative control (FNIII5 at an equimolar concentration).

Our immunoprecipitation data indicated that the 2nd, 4th, 5th and 7th FNIII domains of LAR were the strongest interactors of nidogen-1 and -2 (**Fig 4C, D** of the revised manuscript). To validate this finding in primary neurons, we used a mixture of these domains to block internalisation of the nidogen-H_cT complex. Given that the 1st and 8th FNIII domains also bind to nidogens, albeit not to a great extent, we agree with the Reviewer that fragments FNIII1-4 and FNIII5-8 are likely to be more efficient in out-competing the nidogen-LAR interaction.

We have now combined the nidogen-binding FNIII domains of LAR into two fragments, FNIII1-4-FLAG and FNIII5-7-FLAG, which were tested for their ability to inhibit nidogen-H_cT entry into motor neurons. The 8th FNIII domain, despite being a binding site for nidogens, was not included due to its tendency to cause aggregation.

When these fragments were added together to motor neurons at one-tenth of their binding affinities (0.56 μ M and 0.68 μ M for LAR FNIII1-4-FLAG and FNIII5-7-FLAG, respectively), we found that they were able to halt nidogen-H_cT uptake by ~35% (**Fig R1C, D**). This data is now included in the revised manuscript.

As suggested by this Reviewer, we performed nidogen-H_cT internalisation experiments with equimolar concentrations of LAR FNIII5 domain. As expected, at 1 μ M concentration, there was no change in the uptake of both nidogen-2 and H_cT. However, the data was less clear at higher concentrations (40 and 80 μ M). Under these experimental conditions, nidogen-2 internalisation was unaffected, whilst H_cT uptake was significantly reduced by ~30% (**Fig R5**). A molecular explanation for these results is currently unclear, yet it may be linked to cell toxicity at high protein concentration or non-specific engagement of alternative membrane receptors.

Figure R5. Dose-dependent inhibition of nidogen-2 and HcT-555 uptake in motor neurons by the LAR FNIII5 domain. (A, B) Plots showing internalised nidogen-2 (A) and HcT-555 (B) in motor neurons treated with the indicated concentrations of LAR FNIII5-FLAG. Results were analysed for statistical significance using one-way ANOVA ($P = 0.0417$), followed by Dunnett's multiple comparisons test in (A) and Kruskal-Wallis test ($P = 0.0033$), followed by Dunn's *post-hoc* test in (B). All data are presented as a percentage of internalised nidogen-2 or HcT in buffer-treated controls ($n = 3$ independent experiments; error bars indicate s.e.m.).

3. I wonder if the phosphatase domains of the LAR family protein affect the interaction of LAR with TeNT-nidogen. Does the phosphatase activity have anything to do with the LAR-dependent regulation of TrkB activity?

We thank the Reviewer for raising this point. To address this query, we performed co-immunoprecipitation experiments between full-length LAR and nidogen-2 in N2a cells. We exploited a peptide corresponding to a wedge-shaped, helix-loop-helix region located between the membrane proximal region and the D1 catalytic domain of LAR to inhibit its phosphatase activity (Xie *et al*, 2006). However, using this approach, we were unable to co-immunoprecipitate nidogen-2 with full-length LAR, probably due to the presence of multiple published polymorphisms in the LAR protein.

The interaction data in the present study was acquired using the extracellular domains of LAR and PTPR δ , suggesting that the intracellular phosphatase domain does not play a role in their binding to nidogens. While we agree with the Reviewer that this is an important point that needs experimental validation, we think that it is peripheral to this manuscript and will not change the validity of our current data.

The Longo laboratory had previously shown the BDNF-dependent association between TrkB and the intracellular subunit of LAR (which contains the phosphatase domains), as well as the LAR-dependent regulation of TrkB activity (Yang *et al*, 2006). This study, which was carried out using hippocampal neurons lacking LAR, demonstrated that a decrease in LAR levels was sufficient to downregulate the phosphorylation of the TrkB effectors AKT, ERK and CREB. While the modulation of TrkB activity by LAR was shown to occur *via* the phosphorylation of Src, the authors did not specifically address the role of the phosphatase activity of LAR in this context. Thus, we are unable to posit whether the phosphatase subunit of LAR is responsible for its role in regulating TrkB activity. Nonetheless, under our experimental conditions, blocking TrkB activity using a specific inhibitor and an anti-BDNF antibody did not affect nidogen-HcT uptake, indicating that TrkB activity has no effect on the LAR-nidogen interaction.

4. It is unclear which members of the LAR family proteins are expressed in motor neurons. Approaches such as in situ hybridization may help clarify this issue.

While LAR phosphatases are broadly distributed in the nervous system (Pulido *et al*, 1995; Dunah *et al*, 2005; Kwon *et al*, 2010), it is unclear whether all three proteins are specifically expressed in motor neurons. To clarify this point, we immunostained spinal ventral horn cultures for the three LAR phosphatases as well as the mature motor neuron marker choline acetyl transferase (ChAT) (Barber *et al*, 1984; Sances *et al*, 2016). As expected, we found the presence of all three proteins in ChAT⁺ neurons (**Fig R6**), confirming their expression in spinal cord motor neurons.

Figure R6. Expression of LAR family proteins in primary motor neurons. Representative images of spinal ventral horn cultures stained for the motor neuron marker choline acetyltransferase (ChAT), pan-neuronal marker β III-tubulin and LAR family members. Scale bar: 20 μ m. This is now included in Fig EV1.

5. Presynaptic LAR family proteins are known to interact with several postsynaptic adhesion molecules through various domains. I wonder whether there is any possibility of interference between the known transsynaptic interactions and the FNIII-TeNT/nidogen interaction.

LAR has several trans-synaptic partners, including NGL-3, TrkB, IL1RacP, CSPGs and the HSPGs syndecan and dallylike. The binding sites of TrkB, IL1RacP and CSPGs on LAR are currently unknown (Takahashi & Craig, 2013). Whilst syndecan and dallylike bind to the Ig domains of LAR, NGL-3 interacts with LAR *via* its first two FNIII domains (Johnson *et al*, 2006; Kwon *et al*, 2010). Since the nidogen-TeNT complex also binds to LAR on its FNIII domains, the Reviewer is correct in pointing out that such an interaction might be hindered by a competing LAR-NGL-3 interaction. However, we do not think this is the case. Our truncation and immunoprecipitation analyses have shown that nidogens bind strongly to the individual 2nd, 4th, 5th and 7th FNIII domains of LAR. On the other hand, the 1st and the 8th FNIII domains also bind to nidogens, albeit moderately. Our data also shows that the nidogen-LAR interaction is not dependent on any specific FNIII domain in isolation, but is, in fact, driven by the multivalency of FNIII-nidogen interactions (**Fig 6** of the revised manuscript). This means that even if NGL-3 were to bind to the 2nd FNIII domain, the remaining FNIII sites would still be available for binding to nidogens. This makes the possibility of interference between NGL-3 and nidogen for binding sites on LAR rather unlikely.

6. The LAR panel in Figure 3A should be replaced with a better one.

This panel has now been replaced in Fig 3A, as suggested by the Reviewer.

Appendix.

Sequence alignment of FNIII domains of human PTPR δ and LAR. Black and grey shading denote identical and similar amino acids, respectively.

```

1 .....10.....20.....30.....40.....50.....60
PTPRD 1 VKALPKPPGTPVVTESSTATSITLTWDSGNPEPVSYYIIOHKPKNSEELYKEIDGVATTRY
LAR 1 VKALPKPPIDLVVTEITATSVTLTWDSGNSPEVITYYGIQYRAAGTEGPFQEVLDGVATTRY

61 .....70.....80.....90.....100.....110.....120
PTPRD 61 SVAGLSPMSDYEFRVAVNNIGRGPPSEPVLTQTSEQAPSSAPRDVQARMLSSSTTILVQW
LAR 61 SIGGLSPSEYAFRVAVNSIGRGPPSEAVRARTGEQAPSSPPRRVQARMLSASTMLVQW

121 .....130.....140.....150.....160.....170.....180
PTPRD 121 KEPEEPNGQIQGYRVYYTMDPTQHVNNMVKHNVADSOITTIIGNLVPOKTYSVKVLAFTSI
LAR 121 EPPEEPNGLVRYRVYYTPDSRRPPNAWHKHNTDAGLITTVGSLIPGITYSLRVLAFTAV

181 .....190.....200.....210.....220.....230.....240
PTPRD 181 GDGFLSSDIQVITQTGVPGQPLNFKAEPESETSIILSWTPPRSDTIANYELVYKDGEGHGE
LAR 181 GDGPPSPTIQVKIQGVPAQPADFOAEVESDTRIQLSWLLPQPERIIMYELVYWAAEDED

241 .....250.....260.....270.....280.....290.....300
PTPRD 241 -EQRTIEFPGTSYRLOGLKPNLSLYYFRLAARSPOGIGASTAETISARTMQSKPSAPPQDIS
LAR 241 QQHKVTFDPTSSYTLLEDLKPDI LYRFOLAARSDMGVGVFTPTTEARTAQSTPSAPPQKVM

301 .....310.....320.....330.....340.....350.....360
PTPRD 300 CTSPTSSTSLVSWQPPPVEKONGITEYSIKYTAVDGEDDKPHEILGIPSDTTKYLLEQL
LAR 300 CVSMGSTVVRVSWVPPPADSRNGVITQYSVAHEAVDGEDRGRHVVDGISREHSSWDLVGL

361 .....370.....380.....390.....400.....410.....420
PTPRD 360 EKWTEYRIIVTAHTDVGPGPESLSVLIIRTNEDVPSGPPRKVEVEAVNSTSVKVSWRSPVP
LAR 361 EKWTEYRIVVRAHTDVGPGPESPVLVIRTD EDVPSGPPRKVEVEPLNSTAMHVYWKLVVP

421 .....430.....440.....450.....460.....470.....480
PTPRD 420 NKQHGQIRGYQVHYVRMENGEPKQPMIKDVMLADAQWFFDDTTEHDMIISGLQPETSYS
LAR 421 SKQHGQIRGYQVTVYRIMENGEPKGLPIIQDVMLAEAQWRPEESEDYETTISGLTPETIYS

481 .....490.....500.....510.....520.....530.....540
PTPRD 480 ITVTAYTTKGDGARSKPKLVSTTGAVPGKPRLVINHTQMNTALI QWHPPVDTFGPLQGYR
LAR 481 VTVAAAYTTKGDGARSKPKLVITTGAVPGRPTMMISTTAMNTALI QWHPPKELPGELLGYR

541 .....550.....560.....570.....580.....590.....600
PTPRD 540 LKGRKDMPEPLTTLFSEKEDHFTATDIHKGASYVFRISARNKVGFG EEMVKEISIP EEV
LAR 541 LQVCRAD EARPNTIDFGKDDQHFTVTGLHKGTIYIFRLAAKNRAGLGEEFEKEIRTPEDL

601 .....610.....620.....630.....640.....650.....660
PTPRD 600 PTGFPQNLHSEGTSTSTSVQLSWQPPVLAERNGITTKYTLLYRDINIPLLPMEQLIVPADT
LAR 601 PSGFPQNLHVTGLTSTSTELAWDPPVLAERNGRITISYTVVFRDINSQ---QELQNITTD

661 .....670.....680.....690.....700.....710.....720
PTPRD 660 TMTLLTGLKPDTTYDKVRAHTSKGPGPYSPSVQFRTLVPDQVFAKNFHVAVMKTSVLLS
LAR 658 RFTLLTGLKPDTTYDKVRAWTSKSGPLSPSIQSRTMPVEQVFAKNFRVAAMKTSVLLS

721 .....730.....740.....750.....760.....770.....
PTPRD 720 WEIPEPNYNSAMPFKILYDDGKMVEEVDGRATQKLI VNLKPEKSYSFVLTNRGNSA
LAR 718 WEVPSYKSAVFPFKILYN-GQSV-EVDGHSRMRKLIADLQPNTEYSFVLMNRGS-S

```

Sequence alignment of FNIII domains of human PTPR σ and LAR. Black and grey shading denote identical and similar amino acids, respectively.

```

1 .....10.....20.....30.....40.....50.....60
PTPRS 1 VKSLPKAPGTPMVTENTATSITITWDSGNPDPVSYVYVIEYKSKSQDGPYQIKEDITTRY
LAR 1 VKALPKPPIDLVVTETATSVTLTWDSGNSPEVITYYGIQYRAAGTEGPFQEVGDVATTRY

61 .....70.....80.....90.....100.....110.....120
PTPRS 61 SIGGLSPNSEYEIIVSAVNSIGQGPPESSVVTRTGEQAPASAPRNVQARMLSATTMIVQW
LAR 61 SIGGLSPFSEYAFRVLAVNSIGRGPPEAVRARTGEQAPSSPPRRVQARMLSASTMIVQW

121 .....130.....140.....150.....160.....170.....180
PTPRS 121 EEPVEPNGLRGRVYYTMEPEHPVGNWQKHNVDSDLTTVGSLLEDETYTVRVLAFTSV
LAR 121 EEPVEPNGLRGRVYYTPESSRPPNAWHKHNVDAGLLTTVGSLLPGITYSLRVLAFTAV

181 .....190.....200.....210.....220.....230.....240
PTPRS 181 GDGPLSDPIQVKTOQGVPGQPMNLRAEARSETSTITLSWSPPRQESIITYELLFREG-DHG
LAR 181 GDGPPSPTIQVKTOQGVPAQPADFOAEVESDTRIQLSWLLPPQERIIMYELVYWAAEDED

241 .....250.....260.....270.....280.....290.....300
PTPRS 241 REVGRITFDPTISYVMECLKPNTIYAFRLAARSPQGLGAFTPVVRQRTLQSKPSAPPQDVK
LAR 241 QQHKVITFDPTISYVLECLKPNTIYRFQLAARSDMGVGVFTPTIARTAQSTPSAPPQKVM

301 .....310.....320.....330.....340.....350.....360
PTPRS 300 CVSIRSTALVSWRPPPEPTHNGALVGYSVRYRPIGSEDPEPKVNGIPPTTQILLEAL
LAR 301 CVSMGSTIVRVSWVPPPADSRNGVITQYSVAHEAVDGEDRGRHVVDGISREHSSWDLVGL

361 .....370.....380.....390.....400.....410.....420
PTPRS 360 EKWTQYRITTVAHTEVGGPPESSPVVVRTDEDVPSAPPRKVEAEALNATAIRVLWRSPAP
LAR 361 EKWTQYRIVVRAHTDVGPGPESSPVLVVRTDEDVPSGPPRKVEVEPLNSTAMHVYWKLPVP

421 .....430.....440.....450.....460.....470.....480
PTPRS 421 GFOHGQIRGYQVHYVRMEGAEARPPRIKDVMLADAQ-----EMVITNLQPETAYS
LAR 421 SKOHGQIRGYQVTVYRIENGEPRLPIIQDVMLAEAQWRPEESEDYETTISGLTPETTYS

481 .....490.....500.....510.....520.....530.....540
PTPRS 471 ITVAAAYTKMGDARSKPKVVTTKGAVLGRPTLSVQQTPEGSLARWEPAGTAEDQVLGY
LAR 481 VTVAAAYTKMGDARSKPKIVTTTKGAVPGRPTMMISTAMNTALLQWHPPKEL-PGELLGY

541 .....550.....560.....570.....580.....590.....600
PTPRS 531 RLQIGREDSTPLATLEFPPEEDRYTASGVHKGATYVFRLAARSRGGLGEEAAEVL SIPED
LAR 540 RLQYCRADEARPNTIDFGKDDQHFVIGLHKGTIYIFRLAAKNRAGLGEEFEKEIRTPED

601 .....610.....620.....630.....640.....650.....660
PTPRS 591 TPRGHPOILEAAGNASAGTVLIRWLPPVPAERNGAIVKYTVAVREAGALGPARETELPA
LAR 600 LPSGFPQNLHVTGLTTS--ITELAWDPPVLAERNRIISYTVVFRDINSQQELQ-----

661 .....670.....680.....690.....700.....710.....720
PTPRS 651 AEPGANALITLQGLKPDYAYDLQVRAHTRRGGPFPSPVRYRTFLRDQVSPKNFVKMIM
LAR 652 -NITTDTRFTLITGLKPDITYDIKVRWTSKSGSPLSPSIQSRTMPVEQVFAKNFRVAAAM

721 .....730.....740.....750.....760.....770.....780
PTPRS 711 KTSVLLSWEFPDNYNSPTPYKIQYNGLLTDVDGRTTKKLIITHLKPHTFYNFVLTNRGSS
LAR 711 KTSVLLSWEVPSYKSAVPEKILYNGQSVFVDGHSRKLADIADLPNTEYSFVLMNRGSS-

781 .....790.....800
PTPRS 771 GGLQQTVTAWTAFNLLNGKPS
LAR -----

```

References

- Barber RP, Phelps PE, Houser CR, Crawford GD, Salvaterra PM & Vaughn JE (1984) The morphology and distribution of neurons containing choline acetyltransferase in the adult rat spinal cord: an immunocytochemical study. *J Comp Neurol* 229: 329–346
- Bercsenyi K, Schmiege N, Bryson JB, Wallace M, Caccin P, Golding M, Zanotti G, Greensmith L, Nischt R & Schiavo G (2014) Tetanus toxin entry. Nidogens are therapeutic targets for the prevention of tetanus. *Science* 346: 1118–1123
- Dunah AW, Hueske E, Wyszynski M, Hoogenraad CC, Jaworski J, Pak DT, Simonetta A, Liu G & Sheng M (2005) LAR receptor protein tyrosine phosphatases in the development and maintenance of excitatory synapses. *Nat Neurosci* 8: 458–467
- Johnson KG, Tenney AP, Ghose A, Duckworth AM, Higashi ME, Parfitt K, Marcu O, Heslip TR, Marsh JL, Schwarz TL, *et al* (2006) The HSPGs Syndecan and Dallylike bind the receptor phosphatase LAR and exert distinct effects on synaptic development. *Neuron* 49: 517–531
- Kwon S-K, Woo J, Kim S-Y, Kim H & Kim E (2010) Trans-synaptic adhesions between netrin-G ligand-3 (NGL-3) and receptor tyrosine phosphatases LAR, protein-tyrosine phosphatase delta (PTPdelta), and PTPsigma via specific domains regulate excitatory synapse formation. *J Biol Chem* 285: 13966–13978
- Moritz MS, Tepp WH, Inzalaco HN, Johnson EA & Pellett S (2019) Comparative functional analysis of mice after local injection with botulinum neurotoxin A1, A2, A6, and B1 by catwalk analysis. *Toxicon* 167: 20–28
- Pulido R, Serra-Pagès C, Tang M & Streuli M (1995) The LAR/PTP delta/PTP sigma subfamily of transmembrane protein-tyrosine-phosphatases: multiple human LAR, PTP delta, and PTP sigma isoforms are expressed in a tissue-specific manner and associate with the LAR-interacting protein LIP.1. *Proc Natl Acad Sci USA* 92: 11686–11690
- Sances S, Bruijn LI, Chandran S, Eggan K, Ho R, Klim JR, Livesey MR, Lowry E, Macklis JD, Rushton D, *et al* (2016) Modeling ALS with motor neurons derived from human induced pluripotent stem cells. *Nat Neurosci* 19: 542–553
- Surana S, Tosolini AP, Meyer IFG, Fellows AD, Novoselov SS & Schiavo G (2018) The travel diaries of tetanus and botulinum neurotoxins. *Toxicon* 147: 58–67
- Takahashi H & Craig AM (2013) Protein tyrosine phosphatases PTP δ , PTP σ , and LAR: presynaptic hubs for synapse organization. *Trends Neurosci* 36: 522–534
- Um JW & Ko J (2013) LAR-RPTPs: synaptic adhesion molecules that shape synapse development. *Trends Cell Biol* 23: 465–475
- Xie Y, Massa SM, Ensslen-Craig SE, Major DL, Yang T, Tisi MA, Derevyanny VD, Runge WO, Mehta BP, Moore LA, *et al* (2006) Protein-tyrosine phosphatase (PTP) wedge domain peptides: a novel approach for inhibition of PTP function and augmentation of protein-tyrosine kinase function. *J Biol Chem* 281: 16482–16492
- Yang T, Massa SM & Longo FM (2006) LAR protein tyrosine phosphatase receptor associates with TrkB and modulates neurotrophic signaling pathways. *J Neurobiol* 66: 1420–1436

Dear Gipi, dear Sunaina,

Thank you for the submission of your revised manuscript to The EMBO Journal and your patience. We have now received the comments of both referees who were asked to re-assess your study (included below). As you will see, both referees acknowledge that their initially raised concerns have all been satisfactorily addressed, and they now support publication of the manuscript in our journal.

There are a few minor changes and corrections that we need from you before we can proceed with acceptance of your manuscript:

- Please enter all relevant funding information in our online manuscript handling system. This information should be identical to that provided in the Acknowledgements section of your manuscript (currently missing from the online system: UCL Neurogenetic Therapies Programme funded by The Sigrid Rausing Trust).
- Please consider using our structured methods format for your materials and methods. This section includes a "Reagents and Tools Table" followed by a "Methods and Protocols" section. Please see our guide for a detailed description as well as examples and templates: <https://www.embopress.org/page/journal/14602075/authorguide#researcharticleguide>
- Please note that the Data Availability section is restricted to primary datasets (and computer code) that were generated in the reported study and need deposition in a structured public database (for example, RNA sequencing, mass spectrometry, or structural data). If you have no such datasets to deposit, please state in your Data Availability Section that "This study includes no data deposited in external repositories." The datasets you currently list in your Data Availability statement can instead be provided as Source Data and EV Movies. Our source data coordinator will contact you separately with more information on the requested source data, and how to organize and upload them.
- Please change the heading of your conflict-of-interest statement to "Disclosure and competing interests statement".
- The author contributions statement should be removed from the manuscript file. Instead, we use the CRediT system to specify the contributions of each author in the journal submission system. Please use the free text box to provide more detailed descriptions. See also our guide to authors for more information: <https://www.embopress.org/page/journal/14602075/authorguide#authorshipguidelines>.
- Please note that articles published in our journal typically have no more than 5 EV Figures. We kindly request you to move the remaining 4 EV figures to an Appendix, which should be a single PDF file with a brief Table of Contents with page numbers on its first page. The nomenclature of the Appendix figures should be Appendix Figure S1-S4, and each one of them should be accompanied by its respective legend. Please make sure to update all Figure callouts accordingly throughout your manuscript.
- Please note that the Figures should be removed from the main manuscript file and, instead, only be uploaded as high-resolution individual figure files. Their legends should remain in the main manuscript file, placed below the References list.
- The legend of each movie should be zipped together with the corresponding movie file.
- Please note that EMBO press papers are accompanied online by:
 - A) a short (2 sentences) summary of the findings and their significance,
 - B) 2-5 short bullet points highlighting the key results, and
 - C) a synopsis image in .jpg or .png format that is exactly 550 pixels wide and 300-600 pixels high (the height is variable). Please note that the text needs to be legible at the final size. Please upload this information along with your revised manuscript (the text for A and B should be provided in a separate Word file).

Please also note that as part of the EMBO publications' Transparent Editorial Process, The EMBO Journal publishes online a Peer Review File along with each accepted manuscript. This File will be published in conjunction with your paper and will include the referee reports, your point-by-point response and all pertinent correspondence relating to the manuscript. You can opt out of this by letting the editorial office know (contact@embojournal.org). If you do opt out, the Peer Review File link will point to the following statement: "No Peer Review File is available with this article, as the authors have chosen not to make the review process public in this case."

We look forward to seeing a final version of your manuscript as soon as possible. Please use this link to submit your revision: <https://emboj.msubmit.net/cgi-bin/main.plex>

Best wishes,

Ioannis

Referee #1:

The authors have satisfactorily addressed the reviewers' requests, particularly of in vivo experiments.

Referee #2:

The authors have fully addressed all of my comments. I do not have any additional comments and would like to support the publication of the revised manuscript.

All editorial and formatting issues were resolved by the authors.

Dear Gipi, dear Sunaina,

Congratulations on an excellent manuscript, I am very pleased to inform you that it has been accepted for publication in The EMBO Journal. Thank you for your comprehensive responses to the referee concerns. There are only a few minor formatting changes and textual edits that are still needed before publication, about which I will send you another message shortly.

If you have any questions, please do not hesitate to contact the Editorial Office. Thank you for your contribution to The EMBO Journal. It has been a pleasure working with you!

Best wishes,

Ioannis
